# A Contour Stochastic Gradient Langevin Dynamics Algorithm for Simulations of Multi-modal Distributions

**Wei Deng**
Department of Mathematics
Purdue University
West Lafayette, IN, USA
`weideng056@gmail.com`

**Guang Lin**
Departments of Mathematics &
School of Mechanical Engineering
Purdue University
West Lafayette, IN, USA
`guanglin@purdue.edu`

**Faming Liang** *
Departments of Statistics
Purdue University
West Lafayette, IN, USA
`fmliang@purdue.edu`

## Abstract

We propose an adaptively weighted stochastic gradient Langevin dynamics algorithm (SGLD), so-called contour stochastic gradient Langevin dynamics (CSGLD), for Bayesian learning in big data statistics. The proposed algorithm is essentially a *scalable dynamic importance sampler*, which automatically *flattens* the target distribution such that the simulation for a multi-modal distribution can be greatly facilitated. Theoretically, we prove a stability condition and establish the asymptotic convergence of the self-adapting parameter to a *unique fixed-point*, regardless of the non-convexity of the original energy function; we also present an error analysis for the weighted averaging estimators. Empirically, the CSGLD algorithm is tested on multiple benchmark datasets including CIFAR10 and CIFAR100. The numerical results indicate its superiority over the existing state-of-the-art algorithms in training deep neural networks.

## 1 Introduction

AI safety has long been an important issue in the deep learning community. A promising solution to the problem is Markov chain Monte Carlo (MCMC), which leads to asymptotically correct uncertainty quantification for deep neural network (DNN) models. However, traditional MCMC algorithms [Metropolis et al., 1953, Hastings, 1970] are not scalable to big datasets that deep learning models rely on, although they have achieved significant successes in many scientific areas such as statistical physics and bioinformatics. It was not until the study of stochastic gradient Langevin dynamics (SGLD) [Welling and Teh, 2011] that resolves the scalability issue encountered in Monte Carlo computing for big data problems. Ever since, a variety of scalable stochastic gradient Markov chain Monte Carlo (SGMCMC) algorithms have been developed based on strategies such as Hamiltonian dynamics [Chen et al., 2014, Ma et al., 2015, Ding et al., 2014], Hessian approximation [Ahn et al., 2012, Li et al., 2016, Şimşekli et al., 2016], and higher-order numerical schemes [Chen et al., 2015, Li et al., 2019]. Despite their theoretical guarantees in statistical inference [Chen et al., 2015, Teh et al., 2016, Vollmer et al., 2016] and non-convex optimization [Zhang et al., 2017, Raginsky et al., 2017, Xu et al., 2018], these algorithms often converge slowly, which makes them hard to be used for efficient uncertainty quantification for many AI safety problems.

To develop more efficient SGMCMC algorithms, we seek inspirations from traditional MCMC algorithms, such as simulated annealing [Kirkpatrick et al., 1983], parallel tempering [Swendsen and Wang, 1986, Geyer, 1991], and flat histogram algorithms [Berg and Neuhaus, 1991, Wang and Landau,

2001]. In particular, simulated annealing proposes to decay temperatures to increase the hitting probability to the global optima [Mangoubi and Vishnoi, 2018], which, however, often gets stuck into a local optimum with a fast cooling schedule. Parallel tempering proposes to swap positions of neighboring Markov chains according to an acceptance-rejection rule. However, under the mini-batch setting, it often requires a large correction which is known to deteriorate its performance [Deng et al., 2020a]. The flat histogram algorithms, such as the multicanonical [Berg and Neuhaus, 1991] and Wang-Landau [Wang and Landau, 2001] algorithms, were first proposed to sample discrete states of Ising models by yielding a flat histogram in the energy space, and then extended as a general dynamic importance sampling algorithm, the so-called stochastic approximation Monte Carlo (SAMC) algorithm [Liang, 2005, Liang et al., 2007, Liang, 2009]. Theoretical studies [Lelièvre et al., 2008, Liang, 2010, Fort et al., 2015] support the efficiency of the flat histogram algorithms in Monte Carlo computing for small data problems. However, it is still unclear how to adapt the flat histogram idea to accelerate the convergence of SGMCMC, ensuring efficient uncertainty quantification for AI safety problems.

This paper proposes the so-called contour stochastic gradient Langevin dynamics (CSGLD) algorithm, which successfully extends the flat histogram idea to SGMCMC. Like the SAMC algorithm [Liang, 2005, Liang et al., 2007, Liang, 2009], CSGLD works as a dynamic importance sampling algorithm, which adaptively adjusts the target measure at each iteration and accounts for the bias introduced thereby by importance weights. However, theoretical analysis for the two types of dynamic importance sampling algorithms can be quite different due to the fundamental difference in their transition kernels. We proceed by justifying the stability condition for CSGLD based on the perturbation theory, and establishing ergodicity of CSGLD based on newly developed theory for the convergence of adaptive SGLD. Empirically, we test the performance of CSGLD through a few experiments. It achieves remarkable performance on some synthetic data, UCI datasets, and computer vision datasets such as CIFAR10 and CIFAR100.

## 2  Contour stochastic gradient Langevin dynamics

Suppose we are interested in sampling from a probability measure $\pi(\boldsymbol{x})$ with the density given by

$$\pi(\boldsymbol{x}) \propto \exp(-U(\boldsymbol{x})/\tau), \quad \boldsymbol{x} \in \mathcal{X}, \tag{1}$$

where $\mathcal{X}$ denotes the sample space, $U(\boldsymbol{x})$ is the energy function, and $\tau$ is the temperature. It is known that when $U(\boldsymbol{x})$ is highly non-convex, SGLD can mix very slowly [Raginsky et al., 2017]. To accelerate the convergence, we exploit the flat histogram idea in SGLD.

Suppose that we have partitioned the sample space $\mathcal{X}$ into $m$ subregions based on the energy function $U(\boldsymbol{x})$: $\mathcal{X}_1 = \{\boldsymbol{x} : U(\boldsymbol{x}) \leq u_1\}, \mathcal{X}_2 = \{\boldsymbol{x} : u_1 < U(\boldsymbol{x}) \leq u_2\}, \ldots, \mathcal{X}_{m-1} = \{\boldsymbol{x} : u_{m-2} < U(\boldsymbol{x}) \leq u_{m-1}\}$, and $\mathcal{X}_m = \{\boldsymbol{x} : U(\boldsymbol{x}) > u_{m-1}\}$, where $-\infty < u_1 < u_2 < \cdots < u_{m-1} < \infty$ are specified by the user. For convenience, we set $u_0 = -\infty$ and $u_m = \infty$. Without loss of generality, we assume $u_{i+1} - u_i = \Delta u$ for $i = 1, \ldots, m - 2$. We propose to simulate from a flattened density

$$\varpi_{\Psi_{\boldsymbol{\theta}}}(\boldsymbol{x}) \propto \frac{\pi(\boldsymbol{x})}{\Psi_{\boldsymbol{\theta}}^{\zeta}(U(\boldsymbol{x}))}, \tag{2}$$

where $\zeta > 0$ is a hyperparameter controlling the geometric property of the flatted density (see Figure 1(a) for illustration), and $\boldsymbol{\theta} = (\theta(1), \theta(2), \ldots, \theta(m))$ is an unknown vector taking values in the space:

$$\boldsymbol{\Theta} = \left\{ (\theta(1), \theta(2), \cdots, \theta(m)) \, \middle| \, 0 < \theta(1), \theta(2), \cdots, \theta(m) < 1 \text{ and } \sum_{i=1}^{m} \theta(i) = 1 \right\}. \tag{3}$$

### 2.1  A naïve contour SGLD

It is known if we set [†]

$$\begin{aligned}
&\text{(i) } \zeta = 1 \text{ and } \Psi_{\boldsymbol{\theta}}(U(\boldsymbol{x})) = \sum_{i=1}^{m} \theta(i) 1_{u_{i-1} < U(\boldsymbol{x}) \leq u_i}, \\
&\text{(ii) } \theta(i) = \theta_{\star}(i), \text{ where } \theta_{\star}(i) = \int_{\boldsymbol{\mathcal{X}}_i} \pi(\boldsymbol{x}) d\boldsymbol{x} \text{ for } i \in \{1, 2, \cdots, m\},
\end{aligned} \tag{4}$$

---

[†] $1_A$ is an indicator function that takes value 1 if event $A$ occurs and 0 otherwise.

the algorithm will act like the SAMC algorithm [Liang et al., 2007], yielding a flat histogram in the space of energy (see the pink curve in Fig.1(b)). Theoretically, such a density flattening strategy enables a sharper logarithmic Sobolev inequality and accelerates the convergence of simulations [Lelièvre et al., 2008, Fort et al., 2015]. However, such a density flattening setting only works under the framework of the Metropolis algorithm [Metropolis et al., 1953]. A naïve application of the step function in formula (4(i)) to SGLD results in $\frac{\partial \log \Psi_\theta(u)}{\partial u} = \frac{1}{\Psi_\theta(u)} \frac{\partial \Psi_\theta(u)}{\partial u} = 0$ almost everywhere, which leads to the *vanishing-gradient problem* for SGLD. Calculating the gradient for the naïve contour SGLD, we have

$$\nabla_x \log \varpi_{\Psi_\theta}(x) = - \left[ 1 + \zeta\tau \frac{\partial \log \Psi_\theta(u)}{\partial u} \right] \frac{\nabla_x U(x)}{\tau} = - \frac{\nabla_x U(x)}{\tau}.$$

As such, the naïve algorithm behaves like SGLD and fails to simulate from the flattened density (2).

## 2.2 How to resolve the vanishing gradient

To tackle this issue, we propose to set $\Psi_\theta(u)$ as a piecewise continuous function:

$$\Psi_\theta(u) = \sum_{i=1}^{m} \left( \theta(i-1) e^{(\log \theta(i) - \log \theta(i-1)) \frac{u - u_{i-1}}{\Delta u}} \right) 1_{u_{i-1} < u \leq u_i}, \tag{5}$$

where $\theta(0)$ is fixed to $\theta(1)$ for simplicity. A direct calculation shows that

$$\begin{aligned}
\nabla_x \log \varpi_{\Psi_\theta}(x) &= - \left[ 1 + \zeta\tau \frac{\partial \log \Psi_\theta(u)}{\partial u} \right] \frac{\nabla_x U(x)}{\tau} \\
&= - \left[ 1 + \zeta\tau \frac{\log \theta(J(x)) - \log \theta((J(x) - 1) \vee 1)}{\Delta u} \right] \frac{\nabla_x U(x)}{\tau},
\end{aligned} \tag{6}$$

where $J(x) \in \{1, 2, \cdots, m\}$ denotes the index that $x$ belongs to, i.e., $u_{J(x)-1} < U(x) \leq u_{J(x)}$.[§]

## 2.3 Estimation via stochastic approximation

Since $\theta_\star$ is unknown, we propose to estimate it on the fly under the framework of stochastic approximation [Robbins and Monro, 1951]. Provided that a scalable transition kernel $\Pi_{\theta_k}(x_k, \cdot)$ is available and the energy function $U(x)$ on the full data can be efficiently evaluated, the weighted density $\varpi_{\Psi_\theta}(x)$ can be simulated by iterating between the following steps:

(i) Simulate $x_{k+1}$ from $\Pi_{\theta_k}(x_k, \cdot)$, which admits $\varpi_{\theta_k}(x)$ as the invariant distribution,

(ii) $\theta_{k+1}(i) = \theta_k(i) + \omega_{k+1} \theta_k^\zeta(J(x_{k+1})) \left( 1_{i=J(x_{k+1})} - \theta_k(i) \right)$ for $i \in \{1, 2, \cdots, m\}$. $\qquad(7)$

where $\theta_k$ denotes a working estimate of $\theta$ at the $k$-th iteration. We expect that in a long run, such an algorithm can achieve *an optimization-sampling equilibrium* such that $\theta_k$ converges to the fixed point $\theta_\star$ and the random vector $x_k$ converges weakly to the distribution $\varpi_{\Psi_{\theta_\star}}(x)$.

To make the algorithm scalable to big data, we propose to adopt the Langevin transition kernel for drawing samples at each iteration, for which a mini-batch of data can be used to accelerate computation. In addition, we observe that evaluating $U(x)$ on the full data can be quite expensive for big data problems, while it is free to obtain the stochastic energy $\widetilde{U}(x)$ in evaluating the stochastic gradient $\nabla_x \widetilde{U}(x)$ due to the nature of auto-differentiation [Paszke et al., 2017]. For this reason, we propose a biased index $\tilde{J}(x)$, where $u_{\tilde{J}(x)-1} < \frac{N}{n}\widetilde{U}(x) \leq u_{\tilde{J}(x)}$, $N$ is the sample size of the full dataset and $n$ is the mini-batch size. Let $\{\epsilon_k\}_{k=1}^\infty$ and $\{\omega_k\}_{k=1}^\infty$ denote the learning rates and step sizes for SGLD and stochastic approximation, respectively. Given the above notations, the proposed algorithm can be presented in Algorithm 1, which can be viewed as a *scalable Wang-Landau algorithm* for deep learning and big data problems.

## 2.4 Related work

Compared to the existing MCMC algorithms, the proposed algorithm has a few innovations:

---

[§]Formula (6) shows a practical numerical scheme. An alternative is presented in the supplementary material.

**Algorithm 1** Contour SGLD Algorithm. One can conduct a resampling step from the pool of importance samples according to the importance weights to obtain the original distribution.

---

**[1.] (Data subsampling)** Simulate a mini-batch of data of size $n$ from the whole dataset of size $N$; Compute the stochastic gradient $\nabla_{\boldsymbol{x}} \widetilde{U}(\boldsymbol{x}_k)$ and stochastic energy $\widetilde{U}(\boldsymbol{x}_k)$.

**[2.] (Simulation step)** Sample $\boldsymbol{x}_{k+1}$ using the SGLD algorithm based on $\boldsymbol{x}_k$ and $\boldsymbol{\theta}_k$, i.e.,

$$\boldsymbol{x}_{k+1} = \boldsymbol{x}_k - \epsilon_{k+1} \frac{N}{n} \left[ 1 + \zeta \tau \frac{\log \theta_k(\tilde{J}(\boldsymbol{x}_k)) - \log \theta_k((\tilde{J}(\boldsymbol{x}_k) - 1) \vee 1)}{\Delta u} \right] \nabla_{\boldsymbol{x}} \widetilde{U}(\boldsymbol{x}_k) + \sqrt{2\tau\epsilon_{k+1}} \boldsymbol{w}_{k+1},$$
(8)

where $\boldsymbol{w}_{k+1} \sim N(0, \boldsymbol{I}_d)$, $d$ is the dimension, $\epsilon_{k+1}$ is the learning rate, and $\tau$ is the temperature.

**[3.] (Stochastic approximation)** Update the estimate of $\theta(i)$'s for $i = 1, 2, \ldots, m$ by setting

$$\theta_{k+1}(i) = \theta_k(i) + \omega_{k+1} \theta_k^{\zeta}(\tilde{J}(\boldsymbol{x}_{k+1})) \left( 1_{i=\tilde{J}(\boldsymbol{x}_{k+1})} - \theta_k(i) \right),$$
(9)

where $1_{i=\tilde{J}(\boldsymbol{x}_{k+1})}$ is an indicator function which equals 1 if $i = \tilde{J}(\boldsymbol{x}_{k+1})$ and 0 otherwise.

---

First, CSGLD is an adaptive MCMC algorithm based on the *Langevin transition kernel* instead of the *Metropolis transition kernel* [Liang et al., 2007, Fort et al., 2015]. As a result, the existing convergence theory for the Wang-Landau algorithm does not apply. To resolve this issue, we first prove a stability condition for CSGLD based on the perturbation theory, and then verify regularity conditions for the solution of the Poisson equation so that the fluctuations of the mean-field system induced by CSGLD get controlled, which eventually ensures convergence of CSGLD.

Second, the use of the stochastic index $\tilde{J}(\boldsymbol{x})$ avoids the evaluation of $U(\boldsymbol{x})$ on the full data and thus significantly accelerates the computation of the algorithm, although it leads to a small bias, depending on the mini-batch size $n$, in parameter estimation. Compared to other methods, such as using a fixed sub-dataset to estimate $U(\boldsymbol{x})$, the implementation is much simpler. Moreover, combining the variance reduction of the noisy energy estimators [Deng et al., 2020b], the bias also decreases to zero asymptotically as $\epsilon \to 0$.

Third, unlike the existing SGMCMC algorithms [Welling and Teh, 2011, Chen et al., 2014, Ma et al., 2015], CSGLD works as a *dynamic importance sampler* which *flattens* the target distribution and *reduces the energy barriers* for the sampler to traverse between different regions of the energy landscape (see Fig.1(a) for illustration). The sampling bias introduced thereby is accounted for by the importance weight $\theta^{\zeta}(\tilde{J}(\cdot))$. Interestingly, CSGLD possesses a *self-adjusting mechanism* to ease escapes from local traps, which is similar to the self-repulsive dynamics [Ye et al., 2020] and can be explained as follows. Let's assume that the sampler gets trapped into a local optimum at iteration $k$. Then CSGLD will automatically increase the multiplier of the stochastic gradient (i.e., the bracket term of (8)) at iteration $k + 1$ by increasing the value of $\theta_k(\tilde{J}(\boldsymbol{x}))$, while decreasing the components of $\boldsymbol{\theta}_k$ corresponding to other subregions. This adjustment will continue until the sampler moves away from the current subregion. Then, in the followed several iterations, the multiplier might become negative in neighboring subregions of the local optimum due to the increased value of $\theta(\tilde{J}(\boldsymbol{x}))$, which continues to help to drive the sampler to higher energy regions and thus escape from the local trap. That is, in order to escape from local traps, CSGLD is sometimes forced to move toward higher energy regions by changing the sign of the stochastic gradient multiplier! This is a very attractive feature for simulations of multi-modal distributions.

## 3 Theoretical study of the CSGLD algorithm

In this section, we study the convergence of CSGLD algorithm under the framework of stochastic approximation and show the ergodicity property based on weighted averaging estimators.

### 3.1 Convergence analysis

Following the tradition of stochastic approximation analysis, we rewrite the updating rule (9) as

$$\boldsymbol{\theta}_{k+1} = \boldsymbol{\theta}_k + \omega_{k+1} \widetilde{H}(\boldsymbol{\theta}_k, \boldsymbol{x}_{k+1}),$$
(10)

where $\widetilde{H}(\boldsymbol{\theta}, \boldsymbol{x}) = (\widetilde{H}_1(\boldsymbol{\theta}, \boldsymbol{x}), \ldots, \widetilde{H}_m(\boldsymbol{\theta}, \boldsymbol{x}))$ is a random field function with

$$\widetilde{H}_i(\boldsymbol{\theta}, \boldsymbol{x}) = \theta^\zeta(\tilde{J}(\boldsymbol{x}))\left(1_{i=\tilde{J}(\boldsymbol{x})} - \theta(i)\right), \quad i = 1, 2, \ldots, m. \tag{11}$$

Notably, $\widetilde{H}(\boldsymbol{\theta}, \boldsymbol{x})$ works under an empirical measure $\varpi_{\boldsymbol{\theta}}(\boldsymbol{x})$ which approximates the invariant measure $\varpi_{\Psi_{\boldsymbol{\theta}}}(\boldsymbol{x}) \propto \frac{\pi(\boldsymbol{x})}{\Psi_{\boldsymbol{\theta}}^\zeta(U(\boldsymbol{x}))}$ asymptotically as $\epsilon \to 0$ and $n \to N$. As shown in Lemma 1, we have the mean-field equation

$$h(\boldsymbol{\theta}) = \int_{\mathcal{X}} \widetilde{H}(\boldsymbol{\theta}, \boldsymbol{x})\varpi_{\boldsymbol{\theta}}(\boldsymbol{x})d\boldsymbol{x} = Z_{\boldsymbol{\theta}}^{-1}\left(\boldsymbol{\theta}_\star + \varepsilon\beta(\boldsymbol{\theta}) - \boldsymbol{\theta}\right) = 0, \tag{12}$$

where $\boldsymbol{\theta}_\star = (\int_{\mathcal{X}_1} \pi(\boldsymbol{x})d\boldsymbol{x}, \int_{\mathcal{X}_2} \pi(\boldsymbol{x})d\boldsymbol{x}, \ldots, \int_{\mathcal{X}_m} \pi(\boldsymbol{x})d\boldsymbol{x})$, $Z_{\boldsymbol{\theta}}$ is the normalizing constant, $\beta(\boldsymbol{\theta})$ is a perturbation term, $\varepsilon$ is a small error depending on $\epsilon, n$ and $m$. The mean-field equation implies that for any $\zeta > 0$, $\boldsymbol{\theta}_k$ converges to a small neighbourhood of $\boldsymbol{\theta}_\star$. By applying perturbation theory and setting the Lyapunov function $\mathbb{V}(\boldsymbol{\theta}) = \frac{1}{2}\|\boldsymbol{\theta}_\star - \boldsymbol{\theta}\|^2$, we can establish the stability condition:

**Lemma 1** (Stability). *Given a small enough $\epsilon$ (learning rate), a large enough $n$ (batch size) and $m$ (partition number), there is a constant $\phi = \inf_{\boldsymbol{\theta}} Z_{\boldsymbol{\theta}}^{-1} > 0$ such that the mean-field $h(\boldsymbol{\theta})$ satisfies*

$$\forall \boldsymbol{\theta} \in \boldsymbol{\Theta}, \langle h(\boldsymbol{\theta}), \boldsymbol{\theta} - \boldsymbol{\theta}_\star \rangle \leq -\phi\|\boldsymbol{\theta} - \boldsymbol{\theta}_\star\|^2 + \mathcal{O}\left(\epsilon + \frac{1}{m} + \delta_n(\boldsymbol{\theta})\right),$$

*where $\delta_n(\cdot)$ is a bias term depending on the batch size $n$ and decays to 0 as $n \to N$.*

Together with the tool of Poisson equation [Benveniste et al., 1990, Andrieu et al., 2005], which controls the fluctuation of $\widetilde{H}(\boldsymbol{\theta}, \boldsymbol{x}) - h(\boldsymbol{\theta})$, we can establish convergence of $\boldsymbol{\theta}_k$ in Theorem 1, whose proof is given in the supplementary material.

**Theorem 1** ($L^2$ convergence rate). *Given Assumptions 1-5 (given in Appendix), a small enough learning rate $\epsilon_k$, a large partition number $m$ and a large batch size $n$, $\boldsymbol{\theta}_k$ converges to $\boldsymbol{\theta}_\star$ such that*

$$\mathbb{E}\left[\|\boldsymbol{\theta}_k - \boldsymbol{\theta}_\star\|^2\right] = \mathcal{O}\left(\omega_k + \sup_{i \geq k_0} \epsilon_i + \frac{1}{m} + \sup_{i \geq k_0} \delta_n(\boldsymbol{\theta}_i)\right),$$

*where $k_0$ is some large enough integer, $\boldsymbol{\theta}_\star = (\int_{\mathcal{X}_1} \pi(\boldsymbol{x})d\boldsymbol{x}, \int_{\mathcal{X}_2} \pi(\boldsymbol{x})d\boldsymbol{x}, \ldots, \int_{\mathcal{X}_m} \pi(\boldsymbol{x})d\boldsymbol{x})$, and $\delta_n(\cdot)$ is a bias term depending on the batch size $n$ and decays to 0 as $n \to N$.*

### 3.2 Ergodicity and dynamic importance sampler

CSGLD belongs to the class of adaptive MCMC algorithms, but its transition kernel is based on SGLD instead of the Metropolis algorithm. As such, the ergodicity theory for traditional adaptive MCMC algorithms [Roberts and Rosenthal, 2007, Andrieu and Éric Moulines, 2006, Fort et al., 2011, Liang, 2010] is not directly applicable. To tackle this issue, we conduct the following theoretical study. First, rewrite (8) as

$$\boldsymbol{x}_k - \epsilon\left(\nabla_{\boldsymbol{x}}\widehat{L}(\boldsymbol{x}_k, \boldsymbol{\theta}_\star) + \Upsilon(\boldsymbol{x}_k, \boldsymbol{\theta}_k, \boldsymbol{\theta}_\star)\right) + \mathcal{N}(0, 2\epsilon\tau\boldsymbol{I}), \tag{13}$$

where $\nabla_{\boldsymbol{x}}\widehat{L}(\boldsymbol{x}_k, \boldsymbol{\theta}_\star) = \frac{N}{n}\left[1 + \frac{\zeta\tau}{\Delta u}\left(\log\theta_\star(J(\boldsymbol{x}_k)) - \log\theta_\star((J(\boldsymbol{x}_k) - 1) \vee 1))\right)\right]\nabla_{\boldsymbol{x}}\widetilde{U}(\boldsymbol{x}_k)$, the bias term $\Upsilon(\boldsymbol{x}_k, \boldsymbol{\theta}_k, \boldsymbol{\theta}_\star) = \nabla_{\boldsymbol{x}}\widetilde{L}(\boldsymbol{x}_k, \boldsymbol{\theta}_k) - \nabla_{\boldsymbol{x}}\widehat{L}(\boldsymbol{x}_k, \boldsymbol{\theta}_\star)$, and $\nabla_{\boldsymbol{x}}\widetilde{L}(\boldsymbol{x}_k, \boldsymbol{\theta}_k) = \frac{N}{n}\left[1 + \frac{\zeta\tau}{\Delta u}\left(\log\theta_k(\tilde{J}(\boldsymbol{x}_k)) - \log\theta_k((\tilde{J}(\boldsymbol{x}_k) - 1) \vee 1))\right)\right]\nabla_{\boldsymbol{x}}\widetilde{U}(\boldsymbol{x}_k)$. The order of the bias is figured out in Lemma C1 in the supplementary material based on the results of Theorem 1.

Next, we show how the empirical mean $\frac{1}{k}\sum_{i=1}^{k} f(\boldsymbol{x}_i)$ deviates from the posterior mean $\int_{\mathcal{X}} f(\boldsymbol{x})\varpi_{\Psi_{\boldsymbol{\theta}_\star}}(\boldsymbol{x})d\boldsymbol{x}$. Note that this is a direct application of Theorem 2 of Chen et al. [2015] by treating $\nabla_{\boldsymbol{x}}\widehat{L}(\boldsymbol{x}, \boldsymbol{\theta}_\star)$ as the stochastic gradient of a target distribution and $\Upsilon(\boldsymbol{x}, \boldsymbol{\theta}, \boldsymbol{\theta}_\star)$ as the bias of the stochastic gradient. Moreover, considering that $\varpi_{\widetilde{\Psi}_{\boldsymbol{\theta}_\star}}(\boldsymbol{x}) \propto \frac{\pi(\boldsymbol{x})}{\theta_\star^\zeta(J(\boldsymbol{x}))} \to \varpi_{\Psi_{\boldsymbol{\theta}_\star}}$ as $m \to \infty$ based on Lemma B4 in the supplementary material, we have the following

**Lemma 2** (Convergence of the Averaging Estimators). *Suppose Assumptions 1-6 (in the supplementary material) hold. For any bounded function $f$, we have*

$$\left| \mathbb{E}\left[ \frac{\sum_{i=1}^k f(\boldsymbol{x}_i)}{k} \right] - \int_{\mathcal{X}} f(\boldsymbol{x}) \varpi_{\widetilde{\Psi}_{\boldsymbol{\theta}_\star}}(d\boldsymbol{x}) \right| = \mathcal{O}\left( \frac{1}{k\epsilon} + \sqrt{\epsilon} + \sqrt{\frac{\sum_{i=1}^k \omega_k}{k}} + \frac{1}{\sqrt{m}} + \sup_{i \geq k_0} \sqrt{\delta_n(\boldsymbol{\theta}_i)} \right),$$

*where $\varpi_{\widetilde{\Psi}_{\boldsymbol{\theta}_\star}}(\boldsymbol{x}) = \frac{1}{Z_{\boldsymbol{\theta}_\star}} \frac{\pi(\boldsymbol{x})}{\theta_\star^\zeta(J(\boldsymbol{x}))}$ and $Z_{\boldsymbol{\theta}_\star} = \sum_{i=1}^m \frac{\int_{\mathcal{X}_i} \pi(\boldsymbol{x}) d\boldsymbol{x}}{\theta_\star(i)^\zeta}$.*

Finally, we consider the problem of estimating the quantity $\int_{\mathcal{X}} f(\boldsymbol{x}) \pi(\boldsymbol{x}) d\boldsymbol{x}$. Recall that $\pi(\boldsymbol{x})$ is the target distribution that we would like to make inference for. To estimate this quantity, we naturally consider the weighted averaging estimator $\frac{\sum_{i=1}^k \theta_i^\zeta(\tilde{J}(\boldsymbol{x}_i)) f(\boldsymbol{x}_i)}{\sum_{i=1}^k \theta_i^\zeta(\tilde{J}(\boldsymbol{x}_i))}$ by treating $\theta^\zeta(\tilde{J}(\boldsymbol{x}_i))$ as the dynamic importance weight of the sample $\boldsymbol{x}_i$ for $i = 1, 2, \ldots, k$. The convergence of this estimator is established in Theorem 2, which can be proved by repeated applying Theorem 1 and Lemma 2 with the details given in the supplementary material.

**Theorem 2** (Convergence of the Weighted Averaging Estimators). *Suppose Assumptions 1-6 hold. For any bounded function $f$, we have*

$$\left| \mathbb{E}\left[ \frac{\sum_{i=1}^k \theta_i^\zeta(\tilde{J}(\boldsymbol{x}_i)) f(\boldsymbol{x}_i)}{\sum_{i=1}^k \theta_i^\zeta(\tilde{J}(\boldsymbol{x}_i))} \right] - \int_{\mathcal{X}} f(\boldsymbol{x}) \pi(d\boldsymbol{x}) \right| = \mathcal{O}\left( \frac{1}{k\epsilon} + \sqrt{\epsilon} + \sqrt{\frac{\sum_{i=1}^k \omega_k}{k}} + \frac{1}{\sqrt{m}} + \sup_{i \geq k_0} \sqrt{\delta_n(\boldsymbol{\theta}_i)} \right).$$

The bias of the weighted averaging estimator decreases if one applies a larger batch size, a finer sample space partition, a smaller learning rate $\epsilon$, and smaller step sizes $\{\omega_k\}_{k \geq 0}$. Admittedly, the order of this bias is slightly larger than $\mathcal{O}\left( \frac{1}{k\epsilon} + \epsilon \right)$ achieved by the standard SGLD. We note that this is necessary as simulating from the flattened distribution $\varpi_{\Psi_{\boldsymbol{\theta}_\star}}$ often leads to a much faster convergence, see e.g. the green curve v.s. the purple curve in Fig.1(c).

# 4  Numerical studies

## 4.1  Simulations of multi-modal distributions

**A Gaussian mixture distribution**   The first numerical study is to test the performance of CSGLD on a Gaussian mixture distribution $\pi(\boldsymbol{x}) = 0.4N(-6,1) + 0.6N(4,1)$. In each experiment, the algorithm was run for $10^7$ iterations. We fix the temperature $\tau = 1$ and the learning rate $\epsilon = 0.1$. The step size for stochastic approximation follows $\omega_k = \frac{1}{k^{0.6}+100}$. The sample space is partitioned into 50 subregions with $\Delta u = 1$. The stochastic gradients are simulated by injecting additional random noises following $N(0, 0.01)$ to the exact gradients. For comparison, SGLD is chosen as the baseline algorithm and implemented with the same setup as CSGLD. We repeat the experiments 10 times and report the average and the associated standard deviation.

We first assume that $\boldsymbol{\theta}_\star$ is known and plot the energy functions for both $\pi(\boldsymbol{x})$ and $\varpi_{\Psi_{\boldsymbol{\theta}_\star}}$ with different values of $\zeta$. Fig.1(a) shows that the original energy function has a rather large energy barrier which strongly affects the communication between two modes of the distribution. In contrast, CSGLD samples from a modified energy function, which yields a flattened landscape and reduced energy barriers. For example, with $\zeta = 0.75$, the energy barrier for this example is *greatly reduced from 12 to as small as 2*. Consequently, the local trap problem can be greatly alleviated. Regarding the bizarre peaks around $x = 4$, we leave the study in the supplementary material.

Fig. 1(b) summarizes the estimates of $\boldsymbol{\theta}_\star$ with $\zeta = 0.75$, which matches the ground truth value of $\boldsymbol{\theta}_\star$ very well. Notably, we see that $\theta_\star(i)$ decays exponentially fast as the partition index $i$ increases, which indicates the exponentially decreasing probability of visiting high energy regions and a severe local trap problem. CSGLD tackles this issue by adaptively updating the transition kernel or, equivalently, the invariant distribution such that the sampler moves like a "random walk" in the space of energy. In particular, setting $\zeta = 1$ leads to a flat histogram of energy (for the samples produced by CSGLD).

To explore the performance of CSGLD in quantity estimation with the weighed averaging estimator, we compare CSGLD ($\zeta = 0.75$) with SGLD and KSGLD in estimating the posterior mean $\int_{\mathcal{X}} \boldsymbol{x} \pi(\boldsymbol{x}) d\boldsymbol{x}$, where KSGLD was implemented by assuming $\boldsymbol{\theta}_\star$ is known and sampling from $\varpi_{\Psi_{\boldsymbol{\theta}_\star}}$

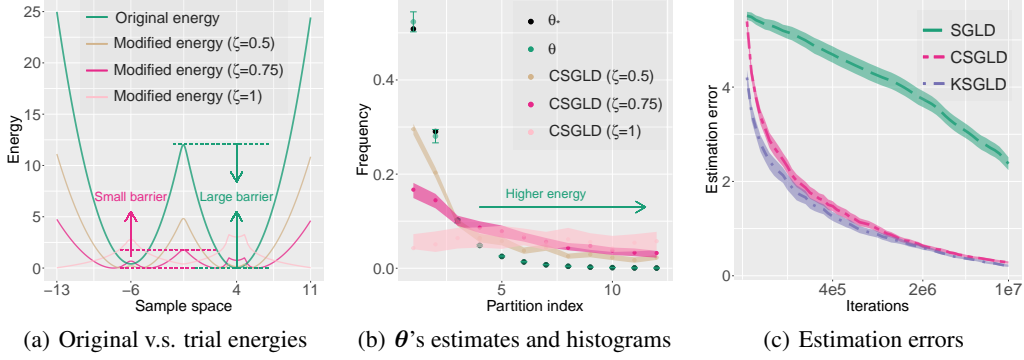

| (a) Original v.s. trial energies | (b) $\boldsymbol{\theta}$'s estimates and histograms | (c) Estimation errors |

Figure 1: Comparison between SGLD and CSGLD: Fig.1(b) presents only the first 12 partitions for an illustrative purpose; KSGLD in Fig.1(c) is implemented by assuming $\boldsymbol{\theta}_\star$ is known.

directly. Each algorithm was run for 10 times, and we recorded the mean absolute estimation error along with iterations. As shown in Fig.1(c), the estimation error of SGLD decays quite slow and rarely converges due to the high energy barrier. On the contrary, KSGLD converges much faster, which shows the advantage of sampling from a flattened distribution $\varpi_{\Psi_{\boldsymbol{\theta}_\star}}$. Admittedly, $\boldsymbol{\theta}_\star$ is unknown in practice. CSGLD instead adaptively updates its invariant distribution while optimizing the parameter $\boldsymbol{\theta}$ until *an optimization-sampling equilibrium* is reached. In the early period of the run, CSGLD converges slightly slower than KSGLD, but soon it becomes as efficient as KSGLD.

Finally, we compare the sample path and learning rate for CSGLD and SGLD. As shown in Fig.2(a), SGLD tends to be trapped in a deep local optimum for an exponentially long time. CSGLD, in contrast, possesses a *self-adjusting mechanism* for escaping from local traps. In the early period of a run, CSGLD might suffer from a similar local-trap problem as SGLD (see Fig.2(b)). In this case, the components of $\boldsymbol{\theta}$ corresponding to the current subregion will increase very fast, eventually rendering a smaller or even negative stochastic gradient multiplier which *bounces the sampler back to high energy regions*. To illustrate the process, we plot a bouncy zone and an absorbing zone in Fig.2(c). The bouncy zone enables the sampler to "jump" over large energy barriers to explore other modes. As the run continues, $\boldsymbol{\theta}_k$ converges to $\boldsymbol{\theta}_\star$. Fig.2(d) shows that larger bouncy "jumps" (in red lines) can potentially be induced in the bouncy zone, which occurs in both local and global optima. Due to the *self-adjusting mechanism*, CSGLD has the local trap problem much alleviated.

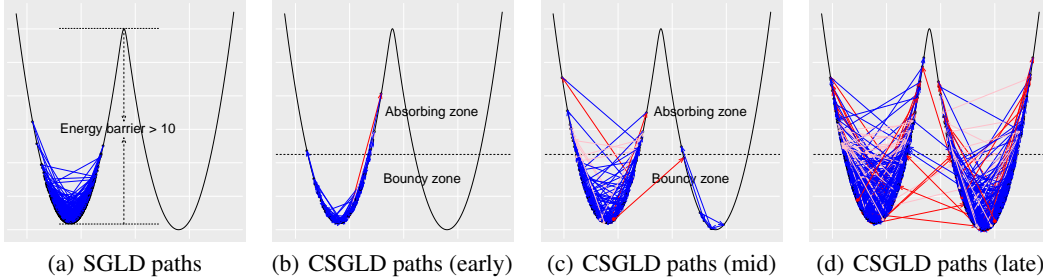

| (a) SGLD paths | (b) CSGLD paths (early) | (c) CSGLD paths (mid) | (d) CSGLD paths (late) |

Figure 2: Sample trajectories of SGLD and CSGLD: plots (a) and (c) are implemented by 100,000 iterations with a thinning factor 100 and $\zeta = 0.75$, while plot (b) utilizes a thinning factor 10.

**A synthetic multi-modal distribution** We next simulate from a distribution $\pi(\boldsymbol{x}) \propto e^{-U(\boldsymbol{x})}$, where $U(\boldsymbol{x}) = \sum_{i=1}^2 \frac{x(i)^2 - 10\cos(1.2\pi x(i))}{3}$ and $\boldsymbol{x} = (x(1), x(2))$. We compare CSGLD with SGLD, replica exchange SGLD (reSGLD) [Deng et al., 2020a], and SGLD with cyclic learning rates (cycSGLD) [Zhang et al., 2020] and detail the setups in the supplementary material. Fig.3(a) shows that the distribution contains nine important modes, where the center mode has the largest probability mass and the four modes on the corners have the smallest mass. We see in Fig.3(b) that SGLD spends too much time in local regions and only identifies three modes. cycSGLD has a better ability to explore the distribution by leveraging large learning rates cyclically. However, as illustrated in Fig.3(c), such a mechanism is still not efficient enough to resolve the local trap issue for this problem. reSGLD proposes to include a high-temperature process to encourage exploration and allows interactions between the two processes via appropriate swaps. We observe in Fig.3(d) that reSGLD obtains both the exploration and exploitation abilities and yields a much better result.

However, the noisy energy estimator may hinder the swapping efficiency and it becomes difficult to estimate a few modes on the corners. As to our algorithm, CSGLD first simulates the importance samples and recovers the original distribution according to the importance weights. We notice that the samples from CSGLD can traverse freely in the parameter space and eventually achieve a remarkable performance, as shown in Fig.3(e).

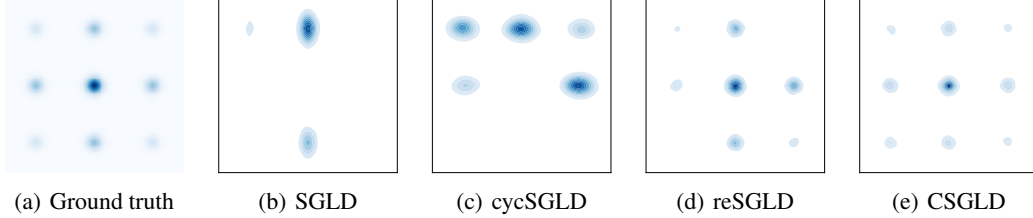

| (a) Ground truth | (b) SGLD | (c) cycSGLD | (d) reSGLD | (e) CSGLD |

Figure 3: Simulations of a multi-modal distribution. A resampling scheme is used for CSGLD.

## 4.2 UCI data

We tested the performance of CSGLD on the **UCI** regression datasets. For each dataset, we normalized all features and randomly selected 10% of the observations for testing. Following [Hernandez-Lobato and Adams, 2015], we modeled the data using a Multi-Layer Perception (MLP) with a single hidden layer of 50 hidden units. We set the mini-batch size $n = 50$ and trained the model for 5,000 epochs. The learning rate was set to 5e-6 and the default $L_2$-regularization coefficient is 1e-4. For all the datasets, we used the stochastic energy $\frac{N}{n}\widetilde{U}(\boldsymbol{x})$ to evaluate the partition index. We set the energy bandwidth $\Delta u = 100$. We fine-tuned the temperature $\tau$ and the hyperparameter $\zeta$. For a fair comparison, each algorithm was run 10 times with fixed seeds for each dataset. In each run, the performance of the algorithm was evaluated by averaging over 50 models, where the averaging estimator was used for SGD and SGLD and the weighted averaging estimator was used for CSGLD. As shown in Table 1, SGLD outperforms the stochastic gradient descent (SGD) algorithm for most datasets due to the advantage of a sampling algorithm in obtaining more informative modes. Since all these datasets are small, there is only very limited potential for improvement. Nevertheless, CSGLD still consistently outperforms all the baselines including SGD and SGLD.

The contour strategy proposed in the paper can be naturally extended to SGHMC [Chen et al., 2014, Ma et al., 2015] without affecting the theoretical results. In what follows, we adopted a numerical method proposed by Saatci and Wilson [2017] to avoid extra hyperparameter tuning. We set the momentum term to 0.9 and simply inherited all the other parameter settings used in the above experiments. In such a case, we compare the contour SGHMC (CSGHMC) with the baselines, including M-SGD (Momentum SGD) and SGHMC. The comparison indicates that some improvements can be achieved by including the momentum.

Table 1: Algorithm evaluation using average root-mean-square error and its standard deviation.

| Dataset<br>Hyperparameters ($\tau/\zeta$) | Energy<br>1/1 | Concrete<br>5/1 | Yacht<br>1/2.5 | Wine<br>5/10 |
|---|---|---|---|---|
| SGD | 1.13±0.07 | 4.60±0.14 | 0.81±0.08 | 0.65±0.01 |
| SGLD | 1.08±0.07 | 4.12±0.10 | 0.72±0.07 | 0.63±0.01 |
| CSGLD | **1.02±0.06** | **3.98±0.11** | **0.69±0.06** | **0.62±0.01** |
| M-SGD | 0.95±0.07 | 4.32±0.27 | 0.73±0.08 | 0.71±0.02 |
| SGHMC | 0.77±0.06 | 4.25±0.19 | **0.66±0.07** | 0.67±0.02 |
| CSGHMC | **0.76±0.06** | **4.15±0.20** | 0.72±0.09 | **0.65±0.01** |

## 4.3 Computer vision data

This section compares only CSGHMC with M-SGD and SGHMC due to the popularity of momentum in accelerating computation for computer vision datasets. We keep partitioning the sample space according to the stochastic energy $\frac{N}{n}\widetilde{U}(\boldsymbol{x})$, where a mini-batch data of size $n$ is randomly chosen from the full dataset of size $N$ at each iteration. Notably, such a strategy significantly accelerates the computation of CSGHMC. As a result, CSGHMC has almost the same computational cost as SGHMC and SGD. To reduce the bias associated with the stochastic energy, we choose a large batch

size $n = 1,000$. For more discussions on the hyperparameter settings, we refer readers to section D in the supplementary material.

**CIFAR10** is a standard computer vision dataset with 10 classes and 60,000 images, for which 50,000 images were used for training and the rest for testing. We modeled the data using a Resnet of 20 layers (Resnet20) [He et al., 2016]. In particular, for CSGHMC, we considered a partition of the energy space in 200 subregions, where the energy bandwidth was set to $\Delta u = 1000$. We trained the model for a total of 1000 epochs and evaluated the model every ten epochs based on two criteria, namely, best point estimate (BPE) and Bayesian model average (BMA). We repeated each experiment 10 times and reported in Table 2 the average prediction accuracy and the corresponding standard deviation.

In the first set of experiments, all the algorithms utilized a fixed learning rate $\epsilon = 2e - 7$ and a fixed temperature $\tau = 0.01$ under the Bayesian setting. SGHMC performs quite similarly to M-SGD, both obtaining around 90% accuracy in BPE and 92% in BMA. Notably, in this case, simulated annealing is not applied to any of the algorithms and achieving the state-of-the-art is quite difficult. However, BMA still consistently outperforms BPE, implying the great potential of advanced MCMC techniques in deep learning. Instead of simulating from $\pi(\boldsymbol{x})$ directly, CSGHMC adaptively simulates from a flattened distribution $\varpi_{\boldsymbol{\theta}_\star}$ and adjusts the sampling bias by dynamic importance weights. As a result, the weighted averaging estimators obtain an improvement by as large as 0.8% on BMA. In addition, the flattened distribution facilitates optimization and the increase in BPE is quite significant.

In the second set of experiments, we employed a decaying schedule on both learning rates and temperatures (if applicable) to obtain simulated annealing effects. For the learning rate, we fix it at $2 \times 10^{-6}$ in the first 400 epochs and then decayed it by a factor of $1.01$ at each epoch. For the temperature, we consistently decayed it by a factor of $1.01$ at each epoch. We call the resulting algorithms by saM-SGD, saSGHMC, and saCSGHMC, respectively. Table 2 shows that the performances of all algorithms are increased quite significantly, where the fine-tuned baselines already obtained the state-of-the-art results. Nevertheless, saCSGHMC further improves BPE by 0.25% and slightly improve the highly optimized BMA by nearly 0.1%.

**CIFAR100** dataset has 100 classes, each of which contains 500 training images and 100 testing images. We follow a similar setup as CIFAR10, except that $\Delta u$ is set to 5000. For M-SGD, BMA can be better than BPE by as large as 5.6%. CSGHMC has led to an improvement of 3.5% on BPE and 2% on BMA, which further demonstrates the superiority of advanced MCMC techniques. Table 2 also shows that with the help of both simulated annealing and importance sampling, saCSGHMC can outperform the highly optimized baselines by almost 1% accuracy on BPE and 0.7% on BMA. The significant improvements show the advantage of the proposed method in training DNNs.

Table 2: Experiments on CIFAR10 & 100 using Resnet20, where BPE and BMA are short for best point estimate and Bayesian model average, respectively.

| Algorithms | CIFAR10 | | CIFAR100 | |
|---|---|---|---|---|
| | BPE | BMA | BPE | BMA |
| M-SGD | 90.02±0.06 | 92.03±0.08 | 61.41±0.15 | 67.04±0.12 |
| SGHMC | 90.01±0.07 | 91.98±0.05 | 61.46±0.14 | 66.43±0.11 |
| CSGHMC | **90.87±0.04** | **92.85±0.05** | **63.97±0.21** | **68.94±0.23** |
| saM-SGD | 93.83±0.07 | 94.25±0.04 | 69.18±0.13 | 71.83±0.12 |
| saSGHMC | 93.80±0.06 | 94.24±0.06 | 69.24±0.11 | 71.98±0.10 |
| saCSGHMC | **94.06±0.07** | 94.33±0.07 | **70.18±0.15** | **72.67±0.15** |

## 5 Conclusion

We have proposed CSGLD as a general scalable Monte Carlo algorithm for both simulation and optimization tasks. CSGLD automatically adjusts the invariant distribution during simulations to facilitate escaping from local traps and traversing over the entire energy landscape. The sampling bias introduced thereby is accounted for by dynamic importance weights. We proved a stability condition for the mean-field system induced by CSGLD together with the convergence of its self-adapting parameter $\boldsymbol{\theta}$ to a unique fixed point $\boldsymbol{\theta}_\star$. We established the convergence of a weighted averaging estimator for CSGLD. The bias of the estimator decreases as we employ a finer partition, a larger mini-batch size, and smaller learning rates and step sizes. We tested CSGLD and its variants on a few examples, which show their great potential in deep learning and big data computing.

## Broader Impact

Our algorithm ensures AI safety by providing more robust predictions and helps build a safer environment. It is an extension of the flat histogram algorithms from the Metropolis kernel to the Langevin kernel and paves the way for future research in various dynamic importance samplers and adaptive biasing force (ABF) techniques for big data problems. The Bayesian community and the researchers in the area of Monte Carlo methods will enjoy the benefit of our work. To our best knowledge, the negative society consequences are not clear and no one will be put at disadvantage.

## Acknowledgment

Liang's research was supported in part by the grants DMS-2015498, R01-GM117597 and R01-GM126089. Lin acknowledges the support from NSF (DMS-1555072, DMS-1736364), BNL Subcontract 382247, W911NF-15-1-0562, and DE-SC0021142.

## Footnotes

*To whom correspondence should be addressed: Faming Liang.

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
