[Supplementary Material]

# Supplementary Material for "A Contour Stochastic Gradient Langevin Dynamics Algorithm for Simulations of Multi-modal Distributions"

**Wei Deng**
Department of Mathematics
Purdue University
West Lafayette, IN, USA
`weideng056@gmail.com`

**Guang Lin**
Departments of Mathematics &
School of Mechanical Engineering
Purdue University
West Lafayette, IN, USA
`guanglin@purdue.edu`

**Faming Liang** *
Departments of Statistics
Purdue University
West Lafayette, IN, USA
`fmliang@purdue.edu`

The supplementary material is organized as follows: Section A provides a review for the related methodologies, Section B proves the stability condition and convergence of the self-adapting parameter, Section C establishes the ergodicity of the contour stochastic gradient Langevin dynamics (CSGLD) algorithm, and Section D provides more discussions for the algorithm.

## A    Background on stochastic approximation and Poisson equation

### A.1    Stochastic approximation

Stochastic approximation [Benveniste et al., 1990] provides a standard framework for the development of adaptive algorithms. Given a random field function $\widetilde{H}(\boldsymbol{\theta}, \boldsymbol{x})$, the goal of the stochastic approximation algorithm is to find the solution to the mean-field equation $h(\boldsymbol{\theta}) = 0$, i.e., solving

$$h(\boldsymbol{\theta}) = \int_{\mathcal{X}} \widetilde{H}(\boldsymbol{\theta}, \boldsymbol{x}) \varpi_{\boldsymbol{\theta}}(d\boldsymbol{x}) = 0,$$

where $\boldsymbol{x} \in \mathcal{X} \subset \mathbb{R}^d$, $\boldsymbol{\theta} \in \Theta \subset \mathbb{R}^m$, $\widetilde{H}(\boldsymbol{\theta}, \boldsymbol{x})$ is a random field function and $\varpi_{\boldsymbol{\theta}}(\boldsymbol{x})$ is a distribution function of $\boldsymbol{x}$ depending on the parameter $\boldsymbol{\theta}$. The stochastic approximation algorithm works by repeating the following iterations

(1) Draw $\boldsymbol{x}_{k+1} \sim \Pi_{\boldsymbol{\theta}_k}(\boldsymbol{x}_k, \cdot)$, where $\Pi_{\boldsymbol{\theta}_k}(\boldsymbol{x}_k, \cdot)$ is a transition kernel that admits $\varpi_{\boldsymbol{\theta}_k}(\boldsymbol{x})$ as the invariant distribution,

(2) Update $\boldsymbol{\theta}_{k+1} = \boldsymbol{\theta}_k + \omega_{k+1}\widetilde{H}(\boldsymbol{\theta}_k, \boldsymbol{x}_{k+1}) + \omega_{k+1}^2 \rho(\boldsymbol{\theta}_k, \boldsymbol{x}_{k+1})$, where $\rho(\cdot, \cdot)$ denotes a bias term.

The algorithm differs from the Robbins–Monro algorithm [Robbins and Monro, 1951] in that $\boldsymbol{x}$ is simulated from a transition kernel $\Pi_{\boldsymbol{\theta}_k}(\cdot, \cdot)$ instead of the exact distribution $\varpi_{\boldsymbol{\theta}_k}(\cdot)$. As a result, a Markov state-dependent noise $\widetilde{H}(\boldsymbol{\theta}_k, \boldsymbol{x}_{k+1}) - h(\boldsymbol{\theta}_k)$ is generated, which requires some regularity conditions to control the fluctuation $\sum_k \Pi_{\boldsymbol{\theta}}^k(\widetilde{H}(\boldsymbol{\theta}, \boldsymbol{x}) - h(\boldsymbol{\theta}))$. Moreover, it supports a more general form where a bounded bias term $\rho(\cdot, \cdot)$ is allowed without affecting the theoretical properties of the algorithm.

### A.2    Poisson equation

Stochastic approximation generates a nonhomogeneous Markov chain $\{(\boldsymbol{x}_k, \boldsymbol{\theta}_k)\}_{k=1}^{\infty}$, for which the convergence theory can be studied based on the Poisson equation

$$\mu_{\boldsymbol{\theta}}(\boldsymbol{x}) - \Pi_{\boldsymbol{\theta}}\mu_{\boldsymbol{\theta}}(\boldsymbol{x}) = \widetilde{H}(\boldsymbol{\theta}, \boldsymbol{x}) - h(\boldsymbol{\theta}),$$

where $\Pi_{\boldsymbol{\theta}}(\boldsymbol{x}, A)$ is the transition kernel for any Borel subset $A \subset \mathcal{X}$ and $\mu_{\boldsymbol{\theta}}(\cdot)$ is a function on $\mathcal{X}$. The solution to the Poisson equation exists when the following series converges:

$$\mu_{\boldsymbol{\theta}}(\boldsymbol{x}) := \sum_{k \geq 0} \Pi_{\boldsymbol{\theta}}^k (\widetilde{H}(\boldsymbol{\theta}, \boldsymbol{x}) - h(\boldsymbol{\theta})).$$

That is, the consistency of the estimator $\boldsymbol{\theta}$ can be established by controlling the perturbations of $\sum_{k \geq 0} \Pi_{\boldsymbol{\theta}}^k (\widetilde{H}(\boldsymbol{\theta}, \boldsymbol{x}) - h(\boldsymbol{\theta}))$ via imposing some regularity conditions on $\mu_{\boldsymbol{\theta}}(\cdot)$. Towards this goal, Benveniste et al. [1990] gave the following regularity conditions on $\mu_{\boldsymbol{\theta}}(\cdot)$ to ensure the convergence of the adaptive algorithm:

*There exist a function $V : \mathcal{X} \rightarrow [1, \infty)$, and a constant $C$ such that for all $\boldsymbol{\theta}, \boldsymbol{\theta}' \in \boldsymbol{\Theta}$,*

$$\|\Pi_{\boldsymbol{\theta}} \mu_{\boldsymbol{\theta}}(\boldsymbol{x})\| \leq CV(\boldsymbol{x}), \quad \|\Pi_{\boldsymbol{\theta}} \mu_{\boldsymbol{\theta}}(\boldsymbol{x}) - \Pi_{\boldsymbol{\theta}'} \mu_{\boldsymbol{\theta}'}(\boldsymbol{x})\| \leq C\|\boldsymbol{\theta} - \boldsymbol{\theta}'\| V(\boldsymbol{x}), \quad \mathbb{E}[V(\boldsymbol{x})] \leq \infty,$$

which requires only the first order smoothness. In contrast, the ergodicity theory by Mattingly et al. [2010] and Vollmer et al. [2016] relies on the much stronger 4th order smoothness.

# B    Stability and convergence analysis for CSGLD

## B.1    CSGLD algorithm

To make the theory more general, we slightly extend CSGLD by allowing a higher order bias term. The resulting algorithm works by iterating between the following two steps:

(1) Sample $\boldsymbol{x}_{k+1} = \boldsymbol{x}_k - \epsilon_k \nabla_{\boldsymbol{x}} \widetilde{L}(\boldsymbol{x}_k, \boldsymbol{\theta}_k) + \mathcal{N}(0, 2\epsilon_k \tau \boldsymbol{I}),$    $(\text{S}_1)$

(2) Update $\boldsymbol{\theta}_{k+1} = \boldsymbol{\theta}_k + \omega_{k+1} \widetilde{H}(\boldsymbol{\theta}_k, \boldsymbol{x}_{k+1}) + \omega_{k+1}^2 \rho(\boldsymbol{\theta}_k, \boldsymbol{x}_{k+1}),$    $(\text{S}_2)$

where $\epsilon_k$ is the learning rate, $\omega_{k+1}$ is the step size, $\nabla_{\boldsymbol{x}} \widetilde{L}(\boldsymbol{x}, \boldsymbol{\theta})$ is the stochastic gradient given by

$$\nabla_{\boldsymbol{x}} \widetilde{L}(\boldsymbol{x}, \boldsymbol{\theta}) = \frac{N}{n} \left[ 1 + \frac{\zeta\tau}{\Delta u} \left( \log \theta(\tilde{J}(\boldsymbol{x})) - \log \theta((\tilde{J}(\boldsymbol{x}) - 1) \vee 1) \right) \right] \nabla_{\boldsymbol{x}} \widetilde{U}(\boldsymbol{x}), \quad (1)$$

$\widetilde{H}(\boldsymbol{\theta}, \boldsymbol{x}) = (\widetilde{H}_1(\boldsymbol{\theta}, \boldsymbol{x}), \ldots, \widetilde{H}_m(\boldsymbol{\theta}, \boldsymbol{x}))$ is a random field function with

$$\widetilde{H}_i(\boldsymbol{\theta}, \boldsymbol{x}) = \theta^\zeta(\tilde{J}(\boldsymbol{x})) \left( 1_{i=\tilde{J}(\boldsymbol{x})} - \theta(i) \right), \quad i = 1, 2, \ldots, m, \quad (2)$$

for some constant $\zeta > 0$, and $\rho(\boldsymbol{\theta}_k, \boldsymbol{x}_{k+1})$ is a bias term.

## B.2    Convergence of parameter estimation

To establish the convergence of $\boldsymbol{\theta}_k$, we make the following assumptions:

**Assumption A1** (Compactness). *The space $\Theta$ is compact such that $\inf_\Theta \theta(i) > 0$ for any $i \in \{1, 2, \ldots, m\}$. There exists a large constant $Q > 0$ such that for any $\boldsymbol{\theta} \in \boldsymbol{\Theta}$ and $\boldsymbol{x} \in \mathcal{X}$,*

$$\|\boldsymbol{\theta}\| \leq Q, \quad \|\widetilde{H}(\boldsymbol{\theta}, \boldsymbol{x})\| \leq Q, \quad \|\rho(\boldsymbol{\theta}, \boldsymbol{x})\| \leq Q. \quad (3)$$

To simplify the proof, we consider a slightly stronger assumption such that $\inf_\Theta \theta(i) > 0$ holds for any $i \in \{1, 2, \ldots, m\}$. To relax this assumption, we refer interested readers to Fort et al. [2015] where the recurrence property was proved for the sequence $\{\boldsymbol{\theta}_k\}_{k \geq 1}$ of a similar algorithm. Such a property guarantees $\boldsymbol{\theta}_k$ to visit often enough to a desired compact space, rendering the convergence of the sequence.

**Assumption A2** (Smoothness). *$U(\boldsymbol{x})$ is $M$-smooth; that is, there exists a constant $M > 0$ such that for any $\boldsymbol{x}, \boldsymbol{x}' \in \mathcal{X}$,*

$$\|\nabla_{\boldsymbol{x}} U(\boldsymbol{x}) - \nabla_{\boldsymbol{x}} U(\boldsymbol{x}')\| \leq M\|\boldsymbol{x} - \boldsymbol{x}'\|. \quad (4)$$

Smoothness is a standard assumption in the study of convergence of SGLD, see e.g. Raginsky et al. [2017], Xu et al. [2018].

**Assumption A3** (Dissipativity). *There exist constants $\tilde{m} > 0$ and $\tilde{b} \geq 0$ such that for any $\boldsymbol{x} \in \mathcal{X}$ and $\boldsymbol{\theta} \in \boldsymbol{\Theta}$,*

$$\langle \nabla_{\boldsymbol{x}} L(\boldsymbol{x}, \boldsymbol{\theta}), \boldsymbol{x} \rangle \leq \tilde{b} - \tilde{m} \|\boldsymbol{x}\|^2. \tag{5}$$

This assumption ensures samples to move towards the origin regardless the initial point, which is standard in proving the geometric ergodicity of dynamical systems, see e.g. Mattingly et al. [2002], Raginsky et al. [2017], Xu et al. [2018].

**Assumption A4** (Gradient noise). *The stochastic gradient is unbiased, that is,*

$$\mathbb{E}[\nabla_{\boldsymbol{x}} \widetilde{U}(\boldsymbol{x}_k) - \nabla_{\boldsymbol{x}} U(\boldsymbol{x}_k)] = 0;$$

*in addition, there exist some constants $M > 0$ and $B > 0$ such that*

$$\mathbb{E}[\|\nabla_{\boldsymbol{x}} \widetilde{U}(\boldsymbol{x}_k) - \nabla_{\boldsymbol{x}} U(\boldsymbol{x}_k)\|^2] \leq M^2 \|\boldsymbol{x}\|^2 + B^2,$$

*where the expectation $\mathbb{E}[\cdot]$ is taken with respect to the distribution of the noise component included in $\nabla_{\boldsymbol{x}} \widetilde{U}(\boldsymbol{x})$.*

Lemma B1 establishes a stability condition for CSGLD, which implies potential convergence of $\boldsymbol{\theta}_k$.

**Lemma B1** (Stability). *Suppose that Assumptions A1-A4 hold. For any $\boldsymbol{\theta} \in \boldsymbol{\Theta}$, $\langle h(\boldsymbol{\theta}), \boldsymbol{\theta} - \boldsymbol{\theta}_\star \rangle \leq -\phi \|\boldsymbol{\theta} - \boldsymbol{\theta}_\star\|^2 + \mathcal{O}\left(\delta_n(\boldsymbol{\theta}) + \epsilon + \frac{1}{m}\right)$, where $\phi = \inf_{\boldsymbol{\theta}} Z_{\boldsymbol{\theta}}^{-1} > 0$, $\boldsymbol{\theta}_\star = \left(\int_{\mathcal{X}_1} \pi(\boldsymbol{x}) d\boldsymbol{x}, \int_{\mathcal{X}_2} \pi(\boldsymbol{x}) d\boldsymbol{x}, \dots, \int_{\mathcal{X}_m} \pi(\boldsymbol{x}) d\boldsymbol{x}\right)$ and $\delta_n(\cdot)$ is a bias term depending on the batch size $n$ such that $\delta_n(\cdot) \to 0$ as $n \to N$.*

**Proof** Let $\varpi_{\Psi_{\boldsymbol{\theta}}}(\boldsymbol{x}) \propto \frac{\pi(\boldsymbol{x})}{\Psi_{\boldsymbol{\theta}}^{\zeta}(U(\boldsymbol{x}))}$ denote a theoretical invariant measure of SGLD, where $\Psi_{\boldsymbol{\theta}}(u)$ is a fixed piecewise continuous function given by

$$\Psi_{\boldsymbol{\theta}}(u) = \sum_{i=1}^{m} \left( \theta(i-1) e^{(\log \theta(i) - \log \theta(i-1)) \frac{u - u_{i-1}}{\Delta u}} \right) 1_{u_{i-1} < u \leq u_i}, \tag{6}$$

the full data is used in determining the indexes of subregions, and the learning rate converges to zero. In addition, we define a piece-wise constant function

$$\widetilde{\Psi}_{\boldsymbol{\theta}} = \sum_{i=1}^{m} \theta(i) 1_{u_{i-1} < u \leq u_i},$$

and a theoretical measure $\varpi_{\widetilde{\Psi}_{\boldsymbol{\theta}}}(\boldsymbol{x}) \propto \frac{\pi(\boldsymbol{x})}{\theta^{\zeta}(J(\boldsymbol{x}))}$. Obviously, as the sample space partition becomes fine and fine, i.e., $u_1 \to u_{\min}$, $u_{m-1} \to u_{\max}$ and $m \to \infty$, we have $\|\widetilde{\Psi}_{\boldsymbol{\theta}} - \Psi_{\boldsymbol{\theta}}\| \to 0$ and $\|\varpi_{\widetilde{\Psi}_{\boldsymbol{\theta}}}(\boldsymbol{x}) - \varpi_{\Psi_{\boldsymbol{\theta}}}(\boldsymbol{x})\| \to 0$, where $u_{\min}$ and $u_{\max}$ denote the minimum and maximum of $U(\boldsymbol{x})$, respectively. Without loss of generality, we assume $u_{\max} < \infty$. Otherwise, $u_{\max}$ can be set to a value such that $\pi(\{\boldsymbol{x} : U(\boldsymbol{x}) > u_{\max}\})$ is sufficiently small.

For each $i \in \{1, 2, \dots, m\}$, the random field $\widetilde{H}_i(\boldsymbol{\theta}, \boldsymbol{x}) = \theta^{\zeta}(\tilde{J}(\boldsymbol{x})) \left(1_{i \geq \tilde{J}(\boldsymbol{x})} - \theta(i)\right)$ is a biased estimator of $H_i(\boldsymbol{\theta}, \boldsymbol{x}) = \theta^{\zeta}(J(\boldsymbol{x})) \left(1_{i \geq J(\boldsymbol{x})} - \theta(i)\right)$. Let $\delta_n(\boldsymbol{\theta}) = \mathbb{E}[\widetilde{H}(\boldsymbol{\theta}, \boldsymbol{x}) - H(\boldsymbol{\theta}, \boldsymbol{x})]$ denote the bias, which is caused by the mini-batch evaluation of the energy and decays to 0 as $n \to N$.

First, let's compute the mean-field $h(\boldsymbol{\theta})$ with respect to the empirical measure $\varpi_{\boldsymbol{\theta}}(\boldsymbol{x})$:

$$h_i(\boldsymbol{\theta}) = \int_{\mathcal{X}} \widetilde{H}_i(\boldsymbol{\theta}, \boldsymbol{x}) \varpi_{\boldsymbol{\theta}}(\boldsymbol{x}) d\boldsymbol{x} = \int_{\mathcal{X}} H_i(\boldsymbol{\theta}, \boldsymbol{x}) \varpi_{\boldsymbol{\theta}}(\boldsymbol{x}) d\boldsymbol{x} + \delta_n(\boldsymbol{\theta})$$

$$= \int_{\mathcal{X}} H_i(\boldsymbol{\theta}, \boldsymbol{x}) \left( \underbrace{\varpi_{\widetilde{\Psi}_{\boldsymbol{\theta}}}(\boldsymbol{x})}_{\text{I}_1} \underbrace{- \varpi_{\widetilde{\Psi}_{\boldsymbol{\theta}}}(\boldsymbol{x}) + \varpi_{\Psi_{\boldsymbol{\theta}}}(\boldsymbol{x})}_{\text{I}_2} \underbrace{- \varpi_{\Psi_{\boldsymbol{\theta}}}(\boldsymbol{x}) + \varpi_{\boldsymbol{\theta}}(\boldsymbol{x})}_{\text{I}_3} \right) d\boldsymbol{x} + \delta_n(\boldsymbol{\theta}). \tag{7}$$

For the term $\text{I}_1$, we have

$$\int_{\mathcal{X}} H_i(\boldsymbol{\theta}, \boldsymbol{x}) \varpi_{\widetilde{\Psi}_{\boldsymbol{\theta}}}(\boldsymbol{x}) d\boldsymbol{x} = \frac{1}{Z_{\boldsymbol{\theta}}} \int_{\mathcal{X}} \theta^{\zeta}(J(\boldsymbol{x})) \left(1_{i=J(\boldsymbol{x})} - \theta(i)\right) \frac{\pi(\boldsymbol{x})}{\theta^{\zeta}(J(\boldsymbol{x}))} d\boldsymbol{x}$$

$$= Z_{\boldsymbol{\theta}}^{-1} \left[ \sum_{k=1}^{m} \int_{\mathcal{X}_k} \pi(\boldsymbol{x}) 1_{k=i} d\boldsymbol{x} - \theta(i) \sum_{k=1}^{m} \int_{\mathcal{X}_k} \pi(\boldsymbol{x}) d\boldsymbol{x} \right] \tag{8}$$

$$= Z_{\boldsymbol{\theta}}^{-1} \left[ \theta_\star(i) - \theta(i) \right],$$

where $Z_{\boldsymbol{\theta}} = \sum_{i=1}^{m} \frac{\int_{\mathcal{X}_i} \pi(\boldsymbol{x}) d\boldsymbol{x}}{\theta(i)^{\varsigma}}$ denotes the normalizing constant of $\varpi_{\widetilde{\Psi}_{\boldsymbol{\theta}}}(\boldsymbol{x})$.

Next, let's consider the integrals $I_2$ and $I_3$. By Lemma B4 and the boundedness of $H(\boldsymbol{\theta}, \boldsymbol{x})$, we have

$$\int_{\mathcal{X}} H_i(\boldsymbol{\theta}, \boldsymbol{x})(-\varpi_{\widetilde{\Psi}_{\boldsymbol{\theta}}}(\boldsymbol{x}) + \varpi_{\Psi_{\boldsymbol{\theta}}}(\boldsymbol{x})) d\boldsymbol{x} = \mathcal{O}\left(\frac{1}{m}\right). \tag{9}$$

For the term $I_3$, we have for any fixed $\boldsymbol{\theta}$,

$$\int_{\mathcal{X}} H_i(\boldsymbol{\theta}, \boldsymbol{x})\left(-\varpi_{\Psi_{\boldsymbol{\theta}}}(\boldsymbol{x}) + \varpi_{\boldsymbol{\theta}}(\boldsymbol{x})\right) d\boldsymbol{x} = \mathcal{O}(\delta_n(\boldsymbol{\theta})) + \mathcal{O}(\epsilon), \tag{10}$$

where $\delta_n(\cdot)$ uniformly decays to 0 as $n \to N$ and the order of $\mathcal{O}(\epsilon)$ follows from Theorem 6 of Sato and Nakagawa [2014].

Plugging (8), (9) and (10) into (7), we have

$$h_i(\boldsymbol{\theta}) = Z_{\boldsymbol{\theta}}^{-1}\left[\varepsilon\beta_i(\boldsymbol{\theta}) + \theta_\star(i) - \theta(i)\right], \tag{11}$$

where $\varepsilon = \mathcal{O}\left(\delta_n(\boldsymbol{\theta}) + \epsilon + \frac{1}{m}\right)$ and $\beta_i(\boldsymbol{\theta})$ is a bounded term such that $Z_{\boldsymbol{\theta}}^{-1}\varepsilon\beta_i(\boldsymbol{\theta}) = \mathcal{O}\left(\delta_n(\boldsymbol{\theta}) + \epsilon + \frac{1}{m}\right)$.

To solve the ODE system with small disturbances, we consider standard techniques in perturbation theory. According to the fundamental theorem of perturbation theory [Vanden-Eijnden, 2001], we can obtain the solution to the mean field equation $h(\boldsymbol{\theta}) = 0$:

$$\theta(i) = \theta_\star(i) + \varepsilon\beta_i(\boldsymbol{\theta}_\star) + \mathcal{O}(\varepsilon^2), \quad i = 1, 2, \ldots, m, \tag{12}$$

which is a stable point in a small neighbourhood of $\boldsymbol{\theta}_\star$.

Considering the positive definite function $\mathbb{V}(\boldsymbol{\theta}) = \frac{1}{2}\|\boldsymbol{\theta}_\star - \boldsymbol{\theta}\|^2$ for the mean-field system $h(\boldsymbol{\theta}) = Z_{\boldsymbol{\theta}}^{-1}(\varepsilon\beta_i(\boldsymbol{\theta}) + \boldsymbol{\theta}_\star - \boldsymbol{\theta}) = Z_{\boldsymbol{\theta}}^{-1}(\boldsymbol{\theta}_\star - \boldsymbol{\theta}) + \mathcal{O}(\varepsilon)$, we have

$$\langle h(\boldsymbol{\theta}), \mathbb{V}(\boldsymbol{\theta})\rangle = \langle h(\boldsymbol{\theta}), \boldsymbol{\theta} - \boldsymbol{\theta}_\star\rangle = -Z_{\boldsymbol{\theta}}^{-1}\|\boldsymbol{\theta} - \boldsymbol{\theta}_\star\|^2 + \mathcal{O}(\varepsilon) \leq -\phi\|\boldsymbol{\theta} - \boldsymbol{\theta}_\star\|^2 + \mathcal{O}\left(\delta_n(\boldsymbol{\theta}) + \epsilon + \frac{1}{m}\right),$$

where $\phi = \inf_{\boldsymbol{\theta}} Z_{\boldsymbol{\theta}}^{-1} > 0$ by the compactness assumption A1. This concludes the proof.

The following is a restatement of Lemma 1 of Deng et al. [2019], which holds for any $\boldsymbol{\theta}$ in the compact space $\boldsymbol{\Theta}$.

**Lemma B2** (Uniform $L^2$ bounds). *Suppose Assumptions A1, A3 and A4 hold. Given a small enough learning rate, then $\sup_{k \geq 1} \mathbb{E}[\|\boldsymbol{x}_k\|^2] < \infty$.*

**Lemma B3** (Solution of Poisson equation). *Suppose that Assumptions A1-A4 hold. There is a solution $\mu_{\boldsymbol{\theta}}(\cdot)$ on $\mathcal{X}$ to the Poisson equation*

$$\mu_{\boldsymbol{\theta}}(\boldsymbol{x}) - \Pi_{\boldsymbol{\theta}}\mu_{\boldsymbol{\theta}}(\boldsymbol{x}) = \widetilde{H}(\boldsymbol{\theta}, \boldsymbol{x}) - h(\boldsymbol{\theta}). \tag{13}$$

*In addition, for all $\boldsymbol{\theta}, \boldsymbol{\theta}' \in \boldsymbol{\Theta}$, there exists a constant $C$ such that*

$$\begin{aligned} \mathbb{E}[\|\Pi_{\boldsymbol{\theta}}\mu_{\boldsymbol{\theta}}(\boldsymbol{x})\|] &\leq C, \\ \mathbb{E}[\|\Pi_{\boldsymbol{\theta}}\mu_{\boldsymbol{\theta}}(\boldsymbol{x}) - \Pi_{\boldsymbol{\theta}'}\mu_{\boldsymbol{\theta}'}(\boldsymbol{x})\|] &\leq C\|\boldsymbol{\theta} - \boldsymbol{\theta}'\|. \end{aligned} \tag{14}$$

**Proof**    The lemma can be proved based on Theorem 13 of Vollmer et al. [2016], whose conditions can be easily verified for CSGLD given the assumptions A1-A4 and Lemma B2. The details are omitted.

Now we are ready to prove the first main result on the convergence of $\boldsymbol{\theta}_k$. The technique lemmas are listed in Section B.3.

**Assumption A5** (Learning rate and step size). *The learning rate $\{\epsilon_k\}_{k \in \mathbb{N}}$ is a positive non-increasing sequence of real numbers satisfying the conditions*

$$\lim_k \epsilon_k = 0, \quad \sum_{k=1}^{\infty} \epsilon_k = \infty.$$

*The step size $\{\omega_k\}_{k \in \mathbb{N}}$ is a positive decreasing sequence of real numbers such that*

$$\omega_k \to 0, \quad \sum_{k=1}^{\infty} \omega_k = +\infty, \quad \lim_{k \to \infty} \inf 2\phi \frac{\omega_k}{\omega_{k+1}} + \frac{\omega_{k+1} - \omega_k}{\omega_{k+1}^2} > 0. \tag{15}$$

*According to Benveniste et al. [1990], we can choose $\omega_k := \frac{A}{k^\alpha + B}$ for some $\alpha \in (\frac{1}{2}, 1]$ and some suitable constants $A > 0$ and $B > 0$.*

**Theorem 1** ($L^2$ convergence rate). *Suppose Assumptions A1-A5 hold. For a sufficiently large value of $m$, a sufficiently small learning rate sequence $\{\epsilon_k\}_{k=1}^{\infty}$, and a sufficiently small step size sequence $\{\omega_k\}_{k=1}^{\infty}$, $\{\boldsymbol{\theta}_k\}_{k=0}^{\infty}$ converges to $\boldsymbol{\theta}_\star$ in $L_2$-norm such that*

$$\mathbb{E}\left[\|\boldsymbol{\theta}_k - \boldsymbol{\theta}_\star\|^2\right] = \mathcal{O}\left(\omega_k + \sup_{i \geq k_0} \epsilon_i + \frac{1}{m} + \sup_{i \geq k_0} \delta_n(\boldsymbol{\theta}_i)\right),$$

*where $k_0$ is a sufficiently large constant, and $\delta_n(\boldsymbol{\theta})$ is a bias term decaying to 0 as $n \to N$.*

**Proof**  Consider the iterations

$$\boldsymbol{\theta}_{k+1} = \boldsymbol{\theta}_k + \omega_{k+1}\left(\widetilde{H}(\boldsymbol{\theta}_k, \boldsymbol{x}_{k+1}) + \omega_{k+1}\rho(\boldsymbol{\theta}_k, \boldsymbol{x}_{k+1})\right).$$

Define $\boldsymbol{T}_k = \boldsymbol{\theta}_k - \boldsymbol{\theta}_\star$. By subtracting $\boldsymbol{\theta}_\star$ from both sides and taking the square and $L_2$ norm, we have

$$\|\boldsymbol{T}_{k+1}^2\| = \|\boldsymbol{T}_k^2\| + \omega_{k+1}^2\|\widetilde{H}(\boldsymbol{\theta}_k, \boldsymbol{x}_{k+1}) + \omega_{k+1}\rho(\boldsymbol{\theta}_k, \boldsymbol{x}_{k+1})\|^2 + 2\omega_{k+1}\underbrace{\langle \boldsymbol{T}_k, \widetilde{H}(\boldsymbol{x}_{k+1}) + \omega_{k+1}\rho(\boldsymbol{\theta}_k, \boldsymbol{x}_{k+1})\rangle}_{\text{D}}.$$

First, by Lemma B5, there exists a constant $G = 4Q^2(1 + Q^2)$ such that

$$\|\widetilde{H}(\boldsymbol{\theta}_k, \boldsymbol{x}_{k+1}) + \omega_{k+1}\rho(\boldsymbol{\theta}_k, \boldsymbol{x}_{k+1})\|^2 \leq G(1 + \|\boldsymbol{T}_k\|^2). \tag{16}$$

Next, by the Poisson equation (13), we have

$$\begin{aligned}
\text{D} &= \langle \boldsymbol{T}_k, \widetilde{H}(\boldsymbol{\theta}_k, \boldsymbol{x}_{k+1}) + \omega_{k+1}\rho(\boldsymbol{\theta}_k, \boldsymbol{x}_{k+1})\rangle \\
&= \langle \boldsymbol{T}_k, h(\boldsymbol{\theta}_k) + \mu_{\boldsymbol{\theta}_k}(\boldsymbol{x}_{k+1}) - \Pi_{\boldsymbol{\theta}_k}\mu_{\boldsymbol{\theta}_k}(\boldsymbol{x}_{k+1}) + \omega_{k+1}\rho(\boldsymbol{\theta}_k, \boldsymbol{x}_{k+1})\rangle \\
&= \underbrace{\langle \boldsymbol{T}_k, h(\boldsymbol{\theta}_k)\rangle}_{\text{D}_1} + \underbrace{\langle \boldsymbol{T}_k, \mu_{\boldsymbol{\theta}_k}(\boldsymbol{x}_{k+1}) - \Pi_{\boldsymbol{\theta}_k}\mu_{\boldsymbol{\theta}_k}(\boldsymbol{x}_{k+1})\rangle}_{\text{D}_2} + \underbrace{\langle \boldsymbol{T}_k, \omega_{k+1}\rho(\boldsymbol{\theta}_k, \boldsymbol{x}_{k+1})\rangle}_{\text{D}_3}.
\end{aligned}$$

For the term $\text{D}_1$, by Lemma B1, we have

$$\mathbb{E}\left[\langle \boldsymbol{T}_k, h(\boldsymbol{\theta}_k)\rangle\right] \leq -\phi\mathbb{E}[\|\boldsymbol{T}_k\|^2] + \mathcal{O}(\delta_n(\boldsymbol{\theta}_k) + \epsilon_k + \frac{1}{m}).$$

For convenience, in the following context, we denote $\mathcal{O}(\delta_n(\boldsymbol{\theta}_k) + \epsilon_k + \frac{1}{m})$ by $\Delta_k$.

To deal with the term $\text{D}_2$, we make the following decomposition

$$\begin{aligned}
\text{D}_2 &= \underbrace{\langle \boldsymbol{T}_k, \mu_{\boldsymbol{\theta}_k}(\boldsymbol{x}_{k+1}) - \Pi_{\boldsymbol{\theta}_k}\mu_{\boldsymbol{\theta}_k}(\boldsymbol{x}_k)\rangle}_{\text{D}_{21}} \\
&\quad + \underbrace{\langle \boldsymbol{T}_k, \Pi_{\boldsymbol{\theta}_k}\mu_{\boldsymbol{\theta}_k}(\boldsymbol{x}_k) - \Pi_{\boldsymbol{\theta}_{k-1}}\mu_{\boldsymbol{\theta}_{k-1}}(\boldsymbol{x}_k)\rangle}_{\text{D}_{22}} + \underbrace{\langle \boldsymbol{T}_k, \Pi_{\boldsymbol{\theta}_{k-1}}\mu_{\boldsymbol{\theta}_{k-1}}(\boldsymbol{x}_k) - \Pi_{\boldsymbol{\theta}_k}\mu_{\boldsymbol{\theta}_k}(\boldsymbol{x}_{k+1})\rangle}_{\text{D}_{23}}.
\end{aligned}$$

(i) From the Markov property, $\mu_{\boldsymbol{\theta}_k}(\boldsymbol{x}_{k+1}) - \Pi_{\boldsymbol{\theta}_k}\mu_{\boldsymbol{\theta}_k}(\boldsymbol{x}_k)$ forms a martingale difference sequence

$$\mathbb{E}\left[\langle \boldsymbol{T}_k, \mu_{\boldsymbol{\theta}_k}(\boldsymbol{x}_{k+1}) - \Pi_{\boldsymbol{\theta}_k}\mu_{\boldsymbol{\theta}_k}(\boldsymbol{x}_k)\rangle | \mathcal{F}_k\right] = 0, \tag{D$_{21}$}$$

where $\mathcal{F}_k$ is a $\sigma$-filter formed by $\{\boldsymbol{\theta}_0, \boldsymbol{x}_1, \boldsymbol{\theta}_1, \boldsymbol{x}_2, \cdots, \boldsymbol{x}_k, \boldsymbol{\theta}_k\}$.

(ii) By the regularity of the solution of Poisson equation in (14) and Lemma B6, we have

$$\mathbb{E}[\|\Pi_{\boldsymbol{\theta}_k}\mu_{\boldsymbol{\theta}_k}(\boldsymbol{x}_k) - \Pi_{\boldsymbol{\theta}_{k-1}}\mu_{\boldsymbol{\theta}_{k-1}}(\boldsymbol{x}_k)\|] \leq C\|\boldsymbol{\theta}_k - \boldsymbol{\theta}_{k-1}\| \leq 2QC\omega_k. \tag{17}$$

Using Cauchy–Schwarz inequality, (17) and the compactness of $\Theta$ in Assumption A1, we have

$$\mathbb{E}[\langle \boldsymbol{T}_k, \Pi_{\boldsymbol{\theta}_k}\mu_{\boldsymbol{\theta}_k}(\boldsymbol{x}_k) - \Pi_{\boldsymbol{\theta}_{k-1}}\mu_{\boldsymbol{\theta}_{k-1}}(\boldsymbol{x}_k)\rangle] \leq \mathbb{E}[\|\boldsymbol{T}_k\|] \cdot 2QC\omega_k \leq 4Q^2C\omega_k \leq 5Q^2C\omega_{k+1} \quad (D_{22}),$$

where the last inequality follows from assumption A5 and holds for a large enough $k$.

(iii) For the last term of $D_2$,

$$\langle \boldsymbol{T}_k, \Pi_{\boldsymbol{\theta}_{k-1}}\mu_{\boldsymbol{\theta}_{k-1}}(\boldsymbol{x}_k) - \Pi_{\boldsymbol{\theta}_k}\mu_{\boldsymbol{\theta}_k}(\boldsymbol{x}_{k+1})\rangle$$
$$= \left(\langle \boldsymbol{T}_k, \Pi_{\boldsymbol{\theta}_{k-1}}\mu_{\boldsymbol{\theta}_{k-1}}(\boldsymbol{x}_k)\rangle - \langle \boldsymbol{T}_{k+1}, \Pi_{\boldsymbol{\theta}_k}\mu_{\boldsymbol{\theta}_k}(\boldsymbol{x}_{k+1})\rangle\right)$$
$$+ \left(\langle \boldsymbol{T}_{k+1}, \Pi_{\boldsymbol{\theta}_k}\mu_{\boldsymbol{\theta}_k}(\boldsymbol{x}_{k+1})\rangle - \langle \boldsymbol{T}_k, \Pi_{\boldsymbol{\theta}_k}\mu_{\boldsymbol{\theta}_k}(\boldsymbol{x}_{k+1})\rangle\right)$$
$$= (z_k - z_{k+1}) + \langle \boldsymbol{T}_{k+1} - \boldsymbol{T}_k, \Pi_{\boldsymbol{\theta}_k}\mu_{\boldsymbol{\theta}_k}(\boldsymbol{x}_{k+1})\rangle,$$

where $z_k = \langle \boldsymbol{T}_k, \Pi_{\boldsymbol{\theta}_{k-1}}\mu_{\boldsymbol{\theta}_{k-1}}(\boldsymbol{x}_k)\rangle$. By the regularity assumption (14) and Lemma B6,

$$\mathbb{E}\langle \boldsymbol{T}_{k+1} - \boldsymbol{T}_k, \Pi_{\boldsymbol{\theta}_k}\mu_{\boldsymbol{\theta}_k}(\boldsymbol{x}_{k+1})\rangle \leq \mathbb{E}[\|\boldsymbol{\theta}_{k+1} - \boldsymbol{\theta}_k\|] \cdot \mathbb{E}[\|\Pi_{\boldsymbol{\theta}_k}\mu_{\boldsymbol{\theta}_k}(\boldsymbol{x}_{k+1})\|] \leq 2QC\omega_{k+1}. \quad (D_{23})$$

Regarding $D_3$, since $\rho(\boldsymbol{\theta}_k, \boldsymbol{x}_{k+1})$ is bounded, applying Cauchy–Schwarz inequality gives

$$\mathbb{E}[\langle \boldsymbol{T}_k, \omega_{k+1}\rho(\boldsymbol{\theta}_k, \boldsymbol{x}_{k+1})\rangle] \leq 2Q^2\omega_{k+1} \quad (D_3)$$

Finally, adding (16), $D_1$, $D_{21}$, $D_{22}$, $D_{23}$ and $D_3$ together, it follows that for a constant $C_0 = G + 10Q^2C + 4QC + 4Q^2$,

$$\mathbb{E}\left[\|\boldsymbol{T}_{k+1}\|^2\right] \leq (1 - 2\omega_{k+1}\phi + G\omega_{k+1}^2)\mathbb{E}\left[\|\boldsymbol{T}_k\|^2\right] + C_0\omega_{k+1}^2 + 2\Delta_k\omega_{k+1} + 2\mathbb{E}[z_k - z_{k+1}]\omega_{k+1}. \tag{18}$$

Moreover, from (3) and (14), $\mathbb{E}[|z_k|]$ is upper bounded by

$$\mathbb{E}[|z_k|] = \mathbb{E}[\langle \boldsymbol{T}_k, \Pi_{\boldsymbol{\theta}_{k-1}}\mu_{\boldsymbol{\theta}_{k-1}}(\boldsymbol{x}_k)\rangle] \leq \mathbb{E}[\|\boldsymbol{T}_k\|]\mathbb{E}[\|\Pi_{\boldsymbol{\theta}_{k-1}}\mu_{\boldsymbol{\theta}_{k-1}}(\boldsymbol{x}_k)\|] \leq 2QC. \tag{19}$$

According to Lemma $B7$, we can choose $\lambda_0$ and $k_0$ such that

$$\mathbb{E}[\|\boldsymbol{T}_{k_0}\|^2] \leq \psi_{k_0} = \lambda_0\omega_{k_0} + \frac{1}{\phi}\sup_{i \geq k_0}\Delta_i,$$

which satisfies the conditions (30) and (31) of Lemma $B9$. Applying Lemma $B9$ leads to

$$\mathbb{E}\left[\|\boldsymbol{T}_k\|^2\right] \leq \psi_k + \mathbb{E}\left[\sum_{j=k_0+1}^{k} \Lambda_j^k (z_{j-1} - z_j)\right], \tag{20}$$

where $\psi_k = \lambda_0\omega_k + \frac{1}{\phi}\sup_{i \geq k_0}\Delta_i$ for all $k > k_0$. Based on (19) and the increasing condition of $\Lambda_j^k$ in Lemma $B8$, we have

$$\mathbb{E}\left[\left|\sum_{j=k_0+1}^{k} \Lambda_j^k (z_{j-1} - z_j)\right|\right] = \mathbb{E}\left[\left|\sum_{j=k_0+1}^{k-1} (\Lambda_{j+1}^k - \Lambda_j^k)z_j - 2\omega_k z_k + \Lambda_{k_0+1}^k z_{k_0}\right|\right]$$
$$\leq \sum_{j=k_0+1}^{k-1} 2(\Lambda_{j+1}^k - \Lambda_j^k)QC + \mathbb{E}[|2\omega_k z_k|] + 2\Lambda_k^k QC \tag{21}$$
$$\leq 2(\Lambda_k^k - \Lambda_{k_0}^k)QC + 2\Lambda_k^k QC + 2\Lambda_k^k QC$$
$$\leq 6\Lambda_k^k QC.$$

Given $\psi_k = \lambda_0\omega_k + \frac{1}{\phi}\sup_{i \geq k_0}\Delta_i$ which satisfies the conditions (30) and (31) of Lemma $B9$, it follows from (20) and (21) that the following inequality holds for any $k > k_0$,

$$\mathbb{E}[\|\boldsymbol{T}_k\|^2] \leq \psi_k + 6\Lambda_k^k QC = (\lambda_0 + 12QC)\,\omega_k + \frac{1}{\phi}\sup_{i \geq k_0}\Delta_i = \lambda\omega_k + \frac{1}{\phi}\sup_{i \geq k_0}\Delta_i,$$

where $\lambda = \lambda_0 + 12QC$, $\lambda_0 = \frac{2G\sup_{i \geq k_0}\Delta_i + 2C_0\phi}{C_1\phi}$, $C_1 = \liminf 2\phi\frac{\omega_k}{\omega_{k+1}} + \frac{\omega_{k+1} - \omega_k}{\omega_{k+1}^2} > 0$, $C_0 = G + 5Q^2C + 2QC + 2Q^2$ and $G = 4Q^2(1 + Q^2)$.

## B.3 Technical lemmas

**Lemma B4.** *Suppose Assumption A1 holds, and $u_1$ and $u_{m-1}$ are fixed such that $\Psi(u_1) > \nu$ and $\Psi(u_{m-1}) > 1 - \nu$ for some small constant $\nu > 0$. For any bounded function $f(\boldsymbol{x})$, we have*

$$\int_{\mathcal{X}} f(\boldsymbol{x}) \left( \varpi_{\Psi_{\boldsymbol{\theta}}}(\boldsymbol{x}) - \varpi_{\widetilde{\Psi}_{\boldsymbol{\theta}}}(\boldsymbol{x}) \right) d\boldsymbol{x} = \mathcal{O}\left(\frac{1}{m}\right). \tag{22}$$

**Proof**  Recall that $\varpi_{\widetilde{\Psi}_{\boldsymbol{\theta}}}(\boldsymbol{x}) = \frac{1}{Z_{\boldsymbol{\theta}}} \frac{\pi(\boldsymbol{x})}{\theta^{\zeta}(J(\boldsymbol{x}))}$ and $\varpi_{\Psi_{\boldsymbol{\theta}}}(\boldsymbol{x}) = \frac{1}{Z_{\Psi_{\boldsymbol{\theta}}}} \frac{\pi(\boldsymbol{x})}{\Psi_{\boldsymbol{\theta}}^{\zeta}(U(\boldsymbol{x}))}$. Since $f(\boldsymbol{x})$ is bounded, it suffices to show

$$\int_{\mathcal{X}} \frac{1}{Z_{\boldsymbol{\theta}}} \frac{\pi(\boldsymbol{x})}{\theta^{\zeta}(J(\boldsymbol{x}))} - \frac{1}{Z_{\Psi_{\boldsymbol{\theta}}}} \frac{\pi(\boldsymbol{x})}{\Psi_{\boldsymbol{\theta}}^{\zeta}(U(\boldsymbol{x}))} d\boldsymbol{x}$$

$$\leq \int_{\mathcal{X}} \left| \frac{1}{Z_{\boldsymbol{\theta}}} \frac{\pi(\boldsymbol{x})}{\theta^{\zeta}(J(\boldsymbol{x}))} - \frac{1}{Z_{\boldsymbol{\theta}}} \frac{\pi(\boldsymbol{x})}{\Psi_{\boldsymbol{\theta}}^{\zeta}(U(\boldsymbol{x}))} \right| d\boldsymbol{x} + \int_{\mathcal{X}} \left| \frac{1}{Z_{\boldsymbol{\theta}}} \frac{\pi(\boldsymbol{x})}{\Psi_{\boldsymbol{\theta}}^{\zeta}(U(\boldsymbol{x}))} - \frac{1}{Z_{\Psi_{\boldsymbol{\theta}}}} \frac{\pi(\boldsymbol{x})}{\Psi_{\boldsymbol{\theta}}^{\zeta}(U(\boldsymbol{x}))} \right| d\boldsymbol{x} \tag{23}$$

$$= \underbrace{\frac{1}{Z_{\boldsymbol{\theta}}} \sum_{i=1}^{m} \int_{\mathcal{X}_i} \left| \frac{\pi(\boldsymbol{x})}{\theta^{\zeta}(i)} - \frac{\pi(\boldsymbol{x})}{\Psi_{\boldsymbol{\theta}}^{\zeta}(U(\boldsymbol{x}))} \right| d\boldsymbol{x}}_{I_1} + \underbrace{\sum_{i=1}^{m} \left| \frac{1}{Z_{\boldsymbol{\theta}}} - \frac{1}{Z_{\Psi_{\boldsymbol{\theta}}}} \right| \int_{\mathcal{X}_i} \frac{\pi(\boldsymbol{x})}{\Psi_{\boldsymbol{\theta}}^{\zeta}(U(\boldsymbol{x}))} d\boldsymbol{x}}_{I_2} = \mathcal{O}\left(\frac{1}{m}\right),$$

where $Z_{\boldsymbol{\theta}} = \sum_{i=1}^{m} \int_{\boldsymbol{X}_i} \frac{\pi(\boldsymbol{x})}{\theta(i)^{\zeta}} d\boldsymbol{x}$, $Z_{\Psi_{\boldsymbol{\theta}}} = \sum_{i=1}^{m} \int_{\boldsymbol{X}_i} \frac{\pi(\boldsymbol{x})}{\Psi_{\boldsymbol{\theta}}^{\zeta}(U(\boldsymbol{x}))} d\boldsymbol{x}$, and $\Psi_{\boldsymbol{\theta}}(u)$ is a piecewise continuous function defined in (6).

By Assumption A1, $\inf_{\Theta} \theta(i) > 0$ for any $i$. Further, by the mean-value theorem, which implies $|x^{\zeta} - y^{\zeta}| \lesssim |x - y| z^{\zeta}$ for any $\zeta > 0, x \leq y$ and $z \in [x, y] \subset [u_1, \infty)$, we have

$$I_1 = \frac{1}{Z_{\boldsymbol{\theta}}} \sum_{i=1}^{m} \int_{\mathcal{X}_i} \left| \frac{\theta^{\zeta}(i) - \Psi_{\boldsymbol{\theta}}^{\zeta}(U(\boldsymbol{x}))}{\theta^{\zeta}(i) \Psi_{\boldsymbol{\theta}}^{\zeta}(U(\boldsymbol{x}))} \right| \pi(\boldsymbol{x}) d\boldsymbol{x} \lesssim \frac{1}{Z_{\boldsymbol{\theta}}} \sum_{i=1}^{m} \int_{\mathcal{X}_i} \frac{|\Psi_{\boldsymbol{\theta}}(u_{i-1}) - \Psi_{\boldsymbol{\theta}}(u_i)|}{\theta^{\zeta}(i)} \pi(\boldsymbol{x}) d\boldsymbol{x}$$

$$\leq \max_i |\Psi_{\boldsymbol{\theta}}(u_i - \Delta u) - \Psi_{\boldsymbol{\theta}}(u_i)| \frac{1}{Z_{\boldsymbol{\theta}}} \sum_{i=1}^{m} \int_{\mathcal{X}_i} \frac{\pi(\boldsymbol{x})}{\theta^{\zeta}(i)} d\boldsymbol{x} = \max_i |\Psi_{\boldsymbol{\theta}}(u_i - \Delta u) - \Psi_{\boldsymbol{\theta}}(u_i)| \lesssim \Delta u = \mathcal{O}\left(\frac{1}{m}\right),$$

where the last inequality follows by Taylor expansion, and the last equality follows as $u_1$ and $u_{m-1}$ are fixed. Similarly, we have

$$I_2 = \left| \frac{1}{Z_{\boldsymbol{\theta}}} - \frac{1}{Z_{\Psi_{\boldsymbol{\theta}}}} \right| Z_{\Psi_{\boldsymbol{\theta}}} = \frac{|Z_{\Psi_{\boldsymbol{\theta}}} - Z_{\boldsymbol{\theta}}|}{Z_{\boldsymbol{\theta}}} \leq \frac{1}{Z_{\boldsymbol{\theta}}} \sum_{i=1}^{m} \int_{\mathcal{X}_i} \left| \frac{\pi(\boldsymbol{x})}{\theta^{\zeta}(i)} - \frac{\pi(\boldsymbol{x})}{\Psi_{\boldsymbol{\theta}}^{\zeta}(U(\boldsymbol{x}))} \right| d\boldsymbol{x} = I_1 = \mathcal{O}\left(\frac{1}{m}\right).$$

The proof can then be concluded by combining the orders of $I_1$ and $I_2$.

**Lemma B5.** *Given $\sup\{\omega_k\}_{k=1}^{\infty} \leq 1$, there exists a constant $G = 4Q^2(1 + Q^2)$ such that*

$$\|\widetilde{H}(\boldsymbol{\theta}_k, \boldsymbol{x}_{k+1}) + \omega_{k+1}\rho(\boldsymbol{\theta}_k, \boldsymbol{x}_{k+1})\|^2 \leq G(1 + \|\boldsymbol{\theta}_k - \boldsymbol{\theta}_\star\|^2). \tag{24}$$

**Proof**

According to the compactness condition in Assumption A1, we have

$$\|\widetilde{H}(\boldsymbol{\theta}_k, \boldsymbol{x}_{k+1})\|^2 \leq Q^2(1 + \|\boldsymbol{\theta}_k\|^2) = Q^2(1 + \|\boldsymbol{\theta}_k - \boldsymbol{\theta}_\star + \boldsymbol{\theta}_\star\|^2) \leq Q^2(1 + 2\|\boldsymbol{\theta}_k - \boldsymbol{\theta}_\star\|^2 + 2Q^2). \tag{25}$$

Therefore, using (25), we can show that for a constant $G = 4Q^2(1 + Q^2)$

$$\|\widetilde{H}(\boldsymbol{\theta}_k, \boldsymbol{x}_{k+1}) + \omega_{k+1}\rho(\boldsymbol{\theta}_k, \boldsymbol{x}_{k+1})\|^2$$
$$\leq 2\|\widetilde{H}(\boldsymbol{\theta}_k, \boldsymbol{x}_{k+1})\|^2 + 2\omega_{k+1}^2 \|\rho(\boldsymbol{\theta}_k, \boldsymbol{x}_{k+1})\|^2$$
$$\leq 2Q^2(1 + 2\|\boldsymbol{\theta}_k - \boldsymbol{\theta}_\star\|^2 + 2Q^2) + 2Q^2$$
$$\leq 2Q^2(2 + 2Q^2 + (2 + 2Q^2)\|\boldsymbol{\theta}_k - \boldsymbol{\theta}_\star\|^2)$$
$$\leq G(1 + \|\boldsymbol{\theta}_k - \boldsymbol{\theta}_\star\|^2).$$

**Lemma B6.** *Given $\sup\{\omega_k\}_{k=1}^{\infty} \leq 1$, we have that*

$$\|\boldsymbol{\theta}_k - \boldsymbol{\theta}_{k-1}\| \leq 2\omega_k Q \tag{26}$$

**Proof**    Following the update $\boldsymbol{\theta}_k - \boldsymbol{\theta}_{k-1} = \omega_k \widetilde{H}(\boldsymbol{\theta}_{k-1}, \boldsymbol{x}_k) + \omega_k^2 \rho(\boldsymbol{\theta}_{k-1}, \boldsymbol{x}_k)$, we have that

$$\|\boldsymbol{\theta}_k - \boldsymbol{\theta}_{k-1}\| = \|\omega_k \widetilde{H}(\boldsymbol{\theta}_{k-1}, \boldsymbol{x}_k) + \omega_k^2 \rho(\boldsymbol{\theta}_{k-1}, \boldsymbol{x}_k)\| \leq \omega_k \|\widetilde{H}(\boldsymbol{\theta}_{k-1}, \boldsymbol{x}_k)\| + \omega_k^2 \|\rho(\boldsymbol{\theta}_{k-1}, \boldsymbol{x}_k)\|.$$

By the compactness condition in Assumption A1 and $\sup\{\omega_k\}_{k=1}^{\infty} \leq 1$, (26) can be derived.

**Lemma B7.** *There exist constants $\lambda_0$ and $k_0$ such that $\forall \lambda \geq \lambda_0$ and $\forall k > k_0$, the sequence $\{\psi_k\}_{k=1}^{\infty}$, where $\psi_k = \lambda \omega_k + \frac{1}{\phi} \sup_{i \geq k_0} \Delta_i$, satisfies*

$$\psi_{k+1} \geq (1 - 2\omega_{k+1}\phi + G\omega_{k+1}^2)\psi_k + C_0\omega_{k+1}^2 + 2\Delta_k\omega_{k+1}. \tag{27}$$

**Proof**    *By replacing $\psi_k$ with $\lambda \omega_k + \frac{1}{\phi} \sup_{i \geq k_0} \Delta_i$ in (27), it suffices to show*

$$\lambda \omega_{k+1} + \frac{1}{\phi} \sup_{i \geq k_0} \Delta_i \geq (1 - 2\omega_{k+1}\phi + G\omega_{k+1}^2) \left( \lambda \omega_k + \frac{1}{\phi} \sup_{i \geq k_0} \Delta_i \right) + C_0\omega_{k+1}^2 + 2\Delta_k\omega_{k+1}.$$

*which is equivalent to proving*

$$\lambda(\omega_{k+1} - \omega_k + 2\omega_k\omega_{k+1}\phi - G\omega_k\omega_{k+1}^2) \geq \frac{1}{\phi} \sup_{i \geq k_0} \Delta_i(-2\omega_{k+1}\phi + G\omega_{k+1}^2) + C_0\omega_{k+1}^2 + 2\Delta_k\omega_{k+1}.$$

*Given the step size condition in (15), we have*

$$\omega_{k+1} - \omega_k + 2\omega_k\omega_{k+1}\phi \geq C_1\omega_{k+1}^2,$$

*where $C_1 = \liminf 2\phi \frac{\omega_k}{\omega_{k+1}} + \frac{\omega_{k+1} - \omega_k}{\omega_{k+1}^2} > 0$. Combining $-\sup_{i \geq k_0} \Delta_i \leq \Delta_k$, it suffices to prove*

$$\lambda (C_1 - G\omega_k) \omega_{k+1}^2 \geq \left( \frac{G}{\phi} \sup_{i \geq k_0} \Delta_i + C_0 \right) \omega_{k+1}^2. \tag{28}$$

*It is clear that for a large enough $k_0$ and $\lambda_0$ such that $\omega_{k_0} \leq \frac{C_1}{2G}$, $\lambda_0 = \frac{2G \sup_{i \geq k_0} \Delta_i + 2C_0\phi}{C_1\phi}$, the desired conclusion (28) holds for all such $k \geq k_0$ and $\lambda \geq \lambda_0$.*

The following lemma is a restatement of Lemma 25 (page 247) from Benveniste et al. [1990].

**Lemma B8.** *Suppose $k_0$ is an integer satisfying $\inf_{k > k_0} \frac{\omega_{k+1} - \omega_k}{\omega_k\omega_{k+1}} + 2\phi - G\omega_{k+1} > 0$ for some constant $G$. Then for any $k > k_0$, the sequence $\{\Lambda_k^K\}_{k=k_0,\ldots,K}$ defined below is increasing and uppered bounded by $2\omega_k$*

$$\Lambda_k^K = \begin{cases} 2\omega_k \prod_{j=k}^{K-1}(1 - 2\omega_{j+1}\phi + G\omega_{j+1}^2) & \text{if } k < K, \\ 2\omega_k & \text{if } k = K. \end{cases} \tag{29}$$

**Lemma B9.** *Let $\{\psi_k\}_{k > k_0}$ be a series that satisfies the following inequality for all $k > k_0$*

$$\psi_{k+1} \geq \psi_k \left( 1 - 2\omega_{k+1}\phi + G\omega_{k+1}^2 \right) + C_0\omega_{k+1}^2 + 2\Delta_k\omega_{k+1}, \tag{30}$$

*and assume there exists such $k_0$ that*

$$\mathbb{E}\left[ \|\boldsymbol{T}_{k_0}\|^2 \right] \leq \psi_{k_0}. \tag{31}$$

*Then for all $k > k_0$, we have*

$$\mathbb{E}\left[ \|\boldsymbol{T}_k\|^2 \right] \leq \psi_k + \sum_{j=k_0+1}^{k} \Lambda_j^k (z_{j-1} - z_j). \tag{32}$$

**Proof**    We prove by the induction method. Assuming (32) is true and applying (18), we have that

$$\mathbb{E}\left[ \|\boldsymbol{T}_{k+1}\|^2 \right] \leq (1 - 2\omega_{k+1}\phi + \omega_{k+1}^2 G)(\psi_k + \sum_{j=k_0+1}^{k} \Lambda_j^k (z_{j-1} - z_j))$$

$$+ C_0\omega_{k+1}^2 + 2\Delta_k\omega_{k+1} + 2\omega_{k+1}\mathbb{E}[z_k - z_{k+1}]$$

Combining (27) and Lemma.B8, respectively, we have

$$\mathbb{E}\left[\|\boldsymbol{T}_{k+1}\|^2\right] \le \psi_{k+1} + (1 - 2\omega_{k+1}\phi + \omega_{k+1}^2 G)\sum_{j=k_0+1}^{k}\Lambda_j^k(z_{j-1} - z_j) + 2\omega_{k+1}\mathbb{E}[z_k - z_{k+1}]$$

$$\le \psi_{k+1} + \sum_{j=k_0+1}^{k}\Lambda_j^{k+1}(z_{j-1} - z_j) + \Lambda_{k+1}^{k+1}\mathbb{E}[z_k - z_{k+1}]$$

$$\le \psi_{k+1} + \sum_{j=k_0+1}^{k+1}\Lambda_j^{k+1}(z_{j-1} - z_j).$$

## C  Ergodicity and dynamic importance sampler

Our interest is to analyze the deviation between the weighted averaging estimator $\frac{1}{k}\sum_{i=1}^{k}\theta_i^\zeta(\tilde{J}(\boldsymbol{x}_i))f(\boldsymbol{x}_i)$ and posterior expectation $\int_{\mathcal{X}} f(\boldsymbol{x})\pi(d\boldsymbol{x})$ for a bounded function $f$. To accomplish this analysis, we first study the convergence of the posterior sample mean $\frac{1}{k}\sum_{i=1}^{k}f(\boldsymbol{x}_i)$ to the posterior expectation $\bar{f} = \int_{\mathcal{X}} f(\boldsymbol{x})\varpi_{\Psi_{\boldsymbol{\theta}_\star}}(\boldsymbol{x})(d\boldsymbol{x})$ and then extend it to $\int_{\mathcal{X}} f(\boldsymbol{x})\varpi_{\widetilde{\Psi}_{\boldsymbol{\theta}_\star}}(\boldsymbol{x})(d\boldsymbol{x})$. The key tool for establishing the ergodic theory is still the Poisson equation which is used to characterize the fluctuation between $f(\boldsymbol{x})$ and $\bar{f}$:

$$\mathcal{L}g(\boldsymbol{x}) = f(\boldsymbol{x}) - \bar{f}, \tag{33}$$

where $g(\boldsymbol{x})$ is the solution of the Poisson equation, and $\mathcal{L}$ is the infinitesimal generator of the Langevin diffusion

$$\mathcal{L}g := \langle\nabla g, \nabla L(\cdot, \boldsymbol{\theta}_\star)\rangle + \tau\nabla^2 g.$$

By imposing the following regularity conditions on the function $g(\boldsymbol{x})$, we can control the perturbations of $\frac{1}{k}\sum_{i=1}^{k}f(\boldsymbol{x}_i) - \bar{f}$ and enables convergence of the weighted averaging estimate.

**Assumption A6** (Regularity). *Given a sufficiently smooth function $g(\boldsymbol{x})$ and a function $\mathcal{V}(\boldsymbol{x})$ such that $\|D^k g\| \lesssim \mathcal{V}^{p_k}(\boldsymbol{x})$ for some constants $p_k > 0$, where $k \in \{0,1,2,3\}$. In addition, $\mathcal{V}^p$ has a bounded expectation, i.e., $\sup_{\boldsymbol{x}}\mathbb{E}[\mathcal{V}^p(\boldsymbol{x})] < \infty$; and $\mathcal{V}$ is smooth, i.e. $\sup_{s\in\{0,1\}}\mathcal{V}^p(s\boldsymbol{x}+(1-s)\boldsymbol{y}) \lesssim \mathcal{V}^p(\boldsymbol{x}) + \mathcal{V}^p(\boldsymbol{y})$ for all $\boldsymbol{x}, \boldsymbol{y} \in \mathcal{X}$ and $p \le 2\max_k\{p_k\}$.*

For stronger but verifiable conditions, we refer readers to Vollmer et al. [2016]. In what follows, we present a lemma, which is majorly adapted from Theorem 2 of Chen et al. [2015] with a fixed learning rate $\epsilon$.

**Lemma C1** (Convergence of the Averaging Estimators). *Suppose Assumptions A1-A6 hold. For any bounded function $f$,*

$$\left|\mathbb{E}\left[\frac{\sum_{i=1}^{k}f(\boldsymbol{x}_i)}{k}\right] - \int_{\mathcal{X}}f(\boldsymbol{x})\varpi_{\widetilde{\Psi}_{\boldsymbol{\theta}_\star}}(\boldsymbol{x})d\boldsymbol{x}\right| = \mathcal{O}\left(\frac{1}{k\epsilon} + \sqrt{\epsilon} + \sqrt{\frac{\sum_{i=1}^{k}\omega_i}{k}} + \frac{1}{\sqrt{m}} + \sup_{i\ge k_0}\sqrt{\delta_n(\boldsymbol{\theta}_i)}\right),$$

*where $k_0$ is a sufficiently large constant, $\varpi_{\widetilde{\Psi}_{\boldsymbol{\theta}_\star}}(\boldsymbol{x}) \propto \frac{\pi(\boldsymbol{x})}{\theta_\star^\zeta(J(\boldsymbol{x}))}$, and $\frac{\sum_{i=1}^{k}\omega_i}{k} = o(\frac{1}{\sqrt{k}})$ as implied by Assumption A5.*

**Proof**  We rewrite the CSGLD algorithm as follows:

$$\boldsymbol{x}_{k+1} = \boldsymbol{x}_k - \epsilon_k\nabla_{\boldsymbol{x}}\widetilde{L}(\boldsymbol{x}_k, \boldsymbol{\theta}_k) + \mathcal{N}(0, 2\epsilon_k\tau\boldsymbol{I})$$

$$= \boldsymbol{x}_k - \epsilon_k\left(\nabla_{\boldsymbol{x}}\widehat{L}(\boldsymbol{x}_k, \boldsymbol{\theta}_\star) + \Upsilon(\boldsymbol{x}_k, \boldsymbol{\theta}_k, \boldsymbol{\theta}_\star)\right) + \mathcal{N}(0, 2\epsilon_k\tau\boldsymbol{I}),$$

where $\nabla_{\boldsymbol{x}}\widehat{L}(\boldsymbol{x}, \boldsymbol{\theta}) = \frac{N}{n}\left[1 + \frac{\zeta\tau}{\Delta u}\left(\log\theta(J(\boldsymbol{x})) - \log\theta((J(\boldsymbol{x}) - 1)\vee 1)\right)\right]\nabla_{\boldsymbol{x}}\widetilde{U}(\boldsymbol{x})$, $\nabla_{\boldsymbol{x}}\widetilde{L}(\boldsymbol{x}, \boldsymbol{\theta})$ is as defined in Section B.1, and the bias term is given by $\Upsilon(\boldsymbol{x}_k, \boldsymbol{\theta}_k, \boldsymbol{\theta}_\star) = \nabla_{\boldsymbol{x}}\widetilde{L}(\boldsymbol{x}_k, \boldsymbol{\theta}_k) - \nabla_{\boldsymbol{x}}\widehat{L}(\boldsymbol{x}_k, \boldsymbol{\theta}_\star)$.

By Assumption A2, we have $\|\nabla_{\boldsymbol{x}}U(\boldsymbol{x})\| = \|\nabla_{\boldsymbol{x}}U(\boldsymbol{x}) - \nabla_{\boldsymbol{x}}U(\boldsymbol{x}_\star)\| \lesssim \|\boldsymbol{x} - \boldsymbol{x}_\star\| \le \|\boldsymbol{x}\| + \|\boldsymbol{x}_\star\|$ for some optimum. Then the $L^2$ upper bound in Lemma B2 implies that $\nabla_{\boldsymbol{x}}U(\boldsymbol{x})$ has a bounded

second moment. Combining Assumption A4, we have $\mathbb{E}\left[\|\nabla_{\boldsymbol{x}}\widetilde{U}(\boldsymbol{x})\|^2\right] < \infty$. Further by Eve's law (i.e., the variance decomposition formula), it is easy to derive that $\mathbb{E}\left[\|\nabla_{\boldsymbol{x}}\widetilde{U}(\boldsymbol{x})\|\right] < \infty$. Then, by the triangle inequality and Jensen's inequality,

$$
\begin{aligned}
\|\mathbb{E}[\Upsilon(\boldsymbol{x}_k, \boldsymbol{\theta}_k, \boldsymbol{\theta}_\star)]\| &\leq \mathbb{E}[\|\nabla_{\boldsymbol{x}}\widetilde{L}(\boldsymbol{x}_k, \boldsymbol{\theta}_k) - \nabla_{\boldsymbol{x}}\widetilde{L}(\boldsymbol{x}_k, \boldsymbol{\theta}_\star)\|] + \mathbb{E}[\|\nabla_{\boldsymbol{x}}\widetilde{L}(\boldsymbol{x}_k, \boldsymbol{\theta}_\star) - \nabla_{\boldsymbol{x}}\widehat{L}(\boldsymbol{x}_k, \boldsymbol{\theta}_\star)\|] \\
&\lesssim \mathbb{E}[\|\boldsymbol{\theta}_k - \boldsymbol{\theta}_\star\|] + \mathcal{O}(\delta_n(\boldsymbol{\theta}_\star)) \leq \sqrt{\mathbb{E}[\|\boldsymbol{\theta}_k - \boldsymbol{\theta}_\star\|^2]} + \mathcal{O}(\delta_n(\boldsymbol{\theta}_\star)) \\
&\leq \mathcal{O}\left(\sqrt{\omega_k + \epsilon + \frac{1}{m} + \sup_{i \geq k_0} \delta_n(\boldsymbol{\theta}_i)}\right),
\end{aligned}
\tag{34}
$$

where Assumption A1 and Theorem 1 are used to derive the smoothness of $\nabla_{\boldsymbol{x}}\widetilde{L}(\boldsymbol{x}, \boldsymbol{\theta})$ with respect to $\boldsymbol{\theta}$, and $\delta_n(\boldsymbol{\theta}) = \mathbb{E}[\widetilde{H}(\boldsymbol{\theta}, \boldsymbol{x}) - H(\boldsymbol{\theta}, \boldsymbol{x})]$ is the bias caused by the mini-batch evaluation of $U(\boldsymbol{x})$.

The ergodic average based on biased gradients and a fixed learning rate is a direct result of Theorem 2 of Chen et al. [2015] by imposing the regularity condition A6. By simulating from $\varpi_{\Psi_{\boldsymbol{\theta}_\star}}(\boldsymbol{x}) \propto \frac{\pi(\boldsymbol{x})}{\Psi_{\boldsymbol{\theta}_\star}^\zeta(U(\boldsymbol{x}))}$ and combining (34) and Theorem 1, we have

$$
\begin{aligned}
\left|\mathbb{E}\left[\frac{\sum_{i=1}^k f(\boldsymbol{x}_i)}{k}\right] - \int_{\mathcal{X}} f(\boldsymbol{x})\varpi_{\Psi_{\boldsymbol{\theta}_\star}}(\boldsymbol{x})d\boldsymbol{x}\right| &\leq \mathcal{O}\left(\frac{1}{k\epsilon} + \epsilon + \frac{\sum_{i=1}^k \|\mathbb{E}[\Upsilon(\boldsymbol{x}_k, \boldsymbol{\theta}_k, \boldsymbol{\theta}_\star)]\|}{k}\right) \\
&\lesssim \mathcal{O}\left(\frac{1}{k\epsilon} + \epsilon + \frac{\sum_{i=1}^k \sqrt{\omega_i + \epsilon + \frac{1}{m} + \sup_{i \geq k_0} \delta_n(\boldsymbol{\theta}_i)}}{k}\right) \\
&\leq \mathcal{O}\left(\frac{1}{k\epsilon} + \sqrt{\epsilon} + \sqrt{\frac{\sum_{i=1}^k \omega_i}{k}} + \frac{1}{\sqrt{m}} + \sup_{i \geq k_0} \sqrt{\delta_n(\boldsymbol{\theta}_i)}\right),
\end{aligned}
$$

where the last inequality follows by repeatedly applying the inequality $\sqrt{a + b} \leq \sqrt{a} + \sqrt{b}$ and the inequality $\sum_{i=1}^k \sqrt{\omega_i} \leq \sqrt{k \sum_{i=1}^k \omega_i}$.

For any a bounded function $f(\boldsymbol{x})$, we have $|\int_{\mathcal{X}} f(\boldsymbol{x})\varpi_{\Psi_{\boldsymbol{\theta}_\star}}(\boldsymbol{x})d\boldsymbol{x} - \int_{\mathcal{X}} f(\boldsymbol{x})\varpi_{\widetilde{\Psi}_{\boldsymbol{\theta}_\star}}(\boldsymbol{x})d\boldsymbol{x}| = \mathcal{O}(\frac{1}{m})$ by Lemma B4. By the triangle inequality, we have

$$
\left|\mathbb{E}\left[\frac{\sum_{i=1}^k f(\boldsymbol{x}_i)}{k}\right] - \int_{\mathcal{X}} f(\boldsymbol{x})\varpi_{\widetilde{\Psi}_{\boldsymbol{\theta}_\star}}(\boldsymbol{x})d\boldsymbol{x}\right| \leq \mathcal{O}\left(\frac{1}{k\epsilon} + \sqrt{\epsilon} + \sqrt{\frac{\sum_{i=1}^k \omega_i}{k}} + \frac{1}{\sqrt{m}} + \sup_{i \geq k_0} \sqrt{\delta(\boldsymbol{\theta}_i)}\right),
$$

which concludes the proof.

Finally, we are ready to show the convergence of the weighted averaging estimator $\frac{\sum_{i=1}^k \theta_i^\zeta(\bar{J}(\boldsymbol{x}_i))f(\boldsymbol{x}_i)}{\sum_{i=1}^k \theta_i^\zeta(\bar{J}(\boldsymbol{x}_i))}$ to the posterior mean $\int_{\mathcal{X}} f(\boldsymbol{x})\pi(d\boldsymbol{x})$.

**Theorem 2** (Convergence of the Weighted Averaging Estimators). *Assume Assumptions A1-A6 hold. For any bounded function $f$, we have that*

$$
\left|\mathbb{E}\left[\frac{\sum_{i=1}^k \theta_i^\zeta(\tilde{J}(\boldsymbol{x}_i))f(\boldsymbol{x}_i)}{\sum_{i=1}^k \theta_i^\zeta(\tilde{J}(\boldsymbol{x}_i))}\right] - \int_{\mathcal{X}} f(\boldsymbol{x})\pi(d\boldsymbol{x})\right| = \mathcal{O}\left(\frac{1}{k\epsilon} + \sqrt{\epsilon} + \sqrt{\frac{\sum_{i=1}^k \omega_i}{k}} + \frac{1}{\sqrt{m}} + \sup_{i \geq k_0} \sqrt{\delta_n(\boldsymbol{\theta}_i)}\right).
$$

**Proof**

Applying triangle inequality and $|\mathbb{E}[x]| \le \mathbb{E}[|x|]$, we have

$$\left| \mathbb{E}\left[ \frac{\sum_{i=1}^{k} \theta_i^{\zeta}(\tilde{J}(\boldsymbol{x}_i)) f(\boldsymbol{x}_i)}{\sum_{i=1}^{k} \theta_i^{\zeta}(\tilde{J}(\boldsymbol{x}_i))} \right] - \int_{\mathcal{X}} f(\boldsymbol{x}) \pi(d\boldsymbol{x}) \right|$$

$$\le \underbrace{\mathbb{E}\left[ \left| \frac{\sum_{i=1}^{k} \theta_i^{\zeta}(\tilde{J}(\boldsymbol{x}_i)) f(\boldsymbol{x}_i)}{\sum_{i=1}^{k} \theta_i^{\zeta}(\tilde{J}(\boldsymbol{x}_i))} - \frac{\sum_{i=1}^{k} \theta_i^{\zeta}(J(\boldsymbol{x}_i)) f(\boldsymbol{x}_i)}{\sum_{i=1}^{k} \theta_i^{\zeta}(J(\boldsymbol{x}_i))} \right| \right]}_{I_1}$$

$$+ \underbrace{\mathbb{E}\left[ \left| \frac{\sum_{i=1}^{k} \theta_i^{\zeta}(J(\boldsymbol{x}_i)) f(\boldsymbol{x}_i)}{\sum_{i=1}^{k} \theta_i^{\zeta}(J(\boldsymbol{x}_i))} - \frac{Z_{\boldsymbol{\theta}_\star} \sum_{i=1}^{k} \theta_i^{\zeta}(J(\boldsymbol{x}_i)) f(\boldsymbol{x}_i)}{k} \right| \right]}_{I_2}$$

$$+ \underbrace{\mathbb{E}\left[ \frac{Z_{\boldsymbol{\theta}_\star}}{k} \sum_{i=1}^{k} \left| \theta_i^{\zeta}(J(\boldsymbol{x}_i)) - \theta_\star^{\zeta}(J(\boldsymbol{x}_i)) \right| \cdot |f(\boldsymbol{x}_i)| \right]}_{I_3} + \underbrace{\left| \mathbb{E}\left[ \frac{Z_{\boldsymbol{\theta}_\star}}{k} \sum_{i=1}^{k} \theta_\star^{\zeta}(J(\boldsymbol{x}_i)) f(\boldsymbol{x}_i) \right] - \int_{\mathcal{X}} f(\boldsymbol{x}) \pi(d\boldsymbol{x}) \right|}_{I_4}.$$

For the term $I_1$, consider the bias $\delta_n(\boldsymbol{\theta}) = \mathbb{E}[\widetilde{H}(\boldsymbol{\theta}, \boldsymbol{x}) - H(\boldsymbol{\theta}, \boldsymbol{x})]$ as defined in the proof of Lemma B1, which decreases to 0 as $n \to N$. By applying mean-value theorem, we have

$$I_1 = \mathbb{E}\left[ \left| \frac{\left( \sum_{i=1}^{k} \theta_i^{\zeta}(\tilde{J}(\boldsymbol{x}_i)) f(\boldsymbol{x}_i) \right) \left( \sum_{i=1}^{k} \theta_i^{\zeta}(J(\boldsymbol{x}_i)) \right) - \left( \sum_{i=1}^{k} \theta_i^{\zeta}(J(\boldsymbol{x}_i)) f(\boldsymbol{x}_i) \right) \left( \sum_{i=1}^{k} \theta_i^{\zeta}(\tilde{J}(\boldsymbol{x}_i)) \right)}{\left( \sum_{i=1}^{k} \theta_i^{\zeta}(\tilde{J}(\boldsymbol{x}_i)) \right) \left( \sum_{i=1}^{k} \theta_i^{\zeta}(J(\boldsymbol{x}_i)) \right)} \right| \right]$$

$$\lesssim \sup_i \delta_n(\boldsymbol{\theta}_i) \mathbb{E}\left[ \frac{\left( \sum_{i=1}^{k} \theta_i^{\zeta}(J(\boldsymbol{x}_i)) f(\boldsymbol{x}_i) \left( \sum_{i=1}^{k} \theta_i^{\zeta}(J(\boldsymbol{x}_i)) \right) \right)}{\left( \sum_{i=1}^{k} \theta_i^{\zeta}(J(\boldsymbol{x}_i)) \right) \left( \sum_{i=1}^{k} \theta_i^{\zeta}(J(\boldsymbol{x}_i)) \right)} \right] = \mathcal{O}\left( \sup_i \delta_n(\boldsymbol{\theta}_i) \right). \tag{35}$$

For the term $I_2$, by the boundedness of $\Theta$ and $f$ and the assumption $\inf_{\Theta} \theta^{\zeta}(i) > 0$, we have

$$I_2 = \mathbb{E}\left[ \left| \frac{\sum_{i=1}^{k} \theta_i^{\zeta}(J(\boldsymbol{x}_i)) f(\boldsymbol{x}_i)}{\sum_{i=1}^{k} \theta_i^{\zeta}(J(\boldsymbol{x}_i))} \left( 1 - \sum_{i=1}^{k} \frac{\theta_i^{\zeta}(J(\boldsymbol{x}_i))}{k} Z_{\boldsymbol{\theta}_\star} \right) \right| \right]$$

$$\lesssim \mathbb{E}\left[ \left| Z_{\boldsymbol{\theta}_\star} \frac{\sum_{i=1}^{k} \theta_i^{\zeta}(J(\boldsymbol{x}_i))}{k} - 1 \right| \right]$$

$$= \mathbb{E}\left[ \left| Z_{\boldsymbol{\theta}_\star} \sum_{i=1}^{m} \frac{\sum_{j=1}^{k} \left( \theta_j^{\zeta}(i) - \theta_\star^{\zeta}(i) + \theta_\star^{\zeta}(i) \right) \mathbb{1}_{J(\boldsymbol{x}_j)=i}}{k} - 1 \right| \right]$$

$$\le \underbrace{\mathbb{E}\left[ Z_{\boldsymbol{\theta}_\star} \sum_{i=1}^{m} \frac{\sum_{j=1}^{k} \left| \theta_j^{\zeta}(i) - \theta_\star^{\zeta}(i) \right| \mathbb{1}_{J(\boldsymbol{x}_j)=i}}{k} \right]}_{I_{21}} + \underbrace{\mathbb{E}\left[ \left| Z_{\boldsymbol{\theta}_\star} \sum_{i=1}^{m} \frac{\theta_\star^{\zeta}(i) \sum_{j=1}^{k} \mathbb{1}_{J(\boldsymbol{x}_j)=i}}{k} - 1 \right| \right]}_{I_{22}}.$$

For $I_{21}$, by first applying the inequality $|x^{\zeta} - y^{\zeta}| \le \zeta |x-y| z^{\zeta-1}$ for any $\zeta > 0$, $x \le y$ and $z \in [x, y]$ based on the mean-value theorem and then applying the Cauchy–Schwarz inequality, we have

$$I_{21} \lesssim \frac{1}{k} \mathbb{E}\left[ \sum_{j=1}^{k} \sum_{i=1}^{m} \left| \theta_j^{\zeta}(i) - \theta_\star^{\zeta}(i) \right| \right] \lesssim \frac{1}{k} \mathbb{E}\left[ \sum_{j=1}^{k} \sum_{i=1}^{m} |\theta_j(i) - \theta_\star(i)| \right] \lesssim \frac{1}{k} \sqrt{\sum_{j=1}^{k} \mathbb{E}\left[ \|\boldsymbol{\theta}_j - \boldsymbol{\theta}_\star\|^2 \right]}, \tag{36}$$

where the compactness of $\Theta$ has been used in deriving the second inequality.

For $I_{22}$, considering the following relation

$$1 = \sum_{i=1}^{m} \int_{\mathcal{X}_i} \pi(\boldsymbol{x}) d\boldsymbol{x} = \sum_{i=1}^{m} \int_{\mathcal{X}_i} \theta_\star^{\zeta}(i) \frac{\pi(\boldsymbol{x})}{\theta_\star^{\zeta}(i)} d\boldsymbol{x} = Z_{\boldsymbol{\theta}_\star} \int_{\mathcal{X}} \sum_{i=1}^{m} \theta_\star^{\zeta}(i) \mathbb{1}_{J(\boldsymbol{x})=i} \varpi_{\tilde{\Psi}_{\boldsymbol{\theta}_\star}}(\boldsymbol{x}) d\boldsymbol{x},$$

then we have

$$
\begin{aligned}
\mathrm{I}_{22} &= \mathbb{E}\left[\left\|Z_{\boldsymbol{\theta}_\star}\sum_{i=1}^{m}\frac{\theta_\star^\zeta(i)\sum_{j=1}^{k}1_{J(\boldsymbol{x}_j)=i}}{k} - Z_{\boldsymbol{\theta}_\star}\int_{\mathcal{X}}\sum_{i=1}^{m}\theta_\star^\zeta(i)1_{J(\boldsymbol{x})=i}\varpi_{\widetilde{\Psi}_{\boldsymbol{\theta}_\star}}(\boldsymbol{x})d\boldsymbol{x}\right\|\right] \\
&= Z_{\boldsymbol{\theta}_\star}\mathbb{E}\left[\left\|\frac{1}{k}\sum_{j=1}^{k}\left(\sum_{i=1}^{m}\theta_\star^\zeta(i)1_{J(\boldsymbol{x}_j)=i}\right) - \int_{\mathcal{X}}\left(\sum_{i=1}^{m}\theta_\star^\zeta(i)1_{J(\boldsymbol{x})=i}\right)\varpi_{\widetilde{\Psi}_{\boldsymbol{\theta}_\star}}(\boldsymbol{x})d\boldsymbol{x}\right\|\right] \qquad (37) \\
&= \mathcal{O}\left(\frac{1}{k\epsilon} + \sqrt{\epsilon} + \sqrt{\frac{\sum_{i=1}^{k}\omega_i}{k}} + \frac{1}{\sqrt{m}} + \sup_{i\geq k_0}\sqrt{\delta_n(\boldsymbol{\theta}_i)}\right),
\end{aligned}
$$

where the last equality follows from Lemma C1 as the step function $\sum_{i=1}^{m}\theta_\star^\zeta(i)1_{J(\boldsymbol{x})=i}$ is integrable. For $\mathrm{I}_3$, by the boundedness of $f$, the mean value theorem and Cauchy-Schwarz inequality, we have

$$
\mathrm{I}_3 \lesssim \mathbb{E}\left[\frac{1}{k}\sum_{i=1}^{k}\left|\theta_i^\zeta(J(\boldsymbol{x}_i)) - \theta_\star^\zeta(J(\boldsymbol{x}_i))\right|\right] \lesssim \frac{1}{k}\mathbb{E}\left[\sum_{j=1}^{k}\sum_{i=1}^{m}|\theta_j(i) - \theta_\star(i)|\right] \lesssim \frac{1}{k}\sqrt{\sum_{j=1}^{k}\mathbb{E}\left[\|\boldsymbol{\theta}_j - \boldsymbol{\theta}_\star\|^2\right]}.
$$

$$(38)$$

For the last term $\mathrm{I}_4$, we first decompose $\int_{\mathcal{X}}f(\boldsymbol{x})\pi(d\boldsymbol{x})$ into $m$ disjoint regions to facilitate the analysis

$$
\begin{aligned}
\int_{\mathcal{X}}f(\boldsymbol{x})\pi(d\boldsymbol{x}) &= \int_{\cup_{j=1}^{m}\mathcal{X}_j}f(\boldsymbol{x})\pi(d\boldsymbol{x}) = \sum_{j=1}^{m}\int_{\mathcal{X}_j}\theta_\star^\zeta(j)f(\boldsymbol{x})\frac{\pi(d\boldsymbol{x})}{\theta_\star^\zeta(j)} \\
&= Z_{\boldsymbol{\theta}_\star}\int_{\mathcal{X}}\sum_{j=1}^{m}\theta_\star(j)^\zeta f(\boldsymbol{x})1_{J(\boldsymbol{x}_i)=j}\varpi_{\widetilde{\Psi}_{\boldsymbol{\theta}_\star}}(\boldsymbol{x})(d\boldsymbol{x}).
\end{aligned}
\qquad (39)
$$

Plugging (39) into the last term $\mathrm{I}_4$, we have

$$
\begin{aligned}
\mathrm{I}_4 &= \left|\mathbb{E}\left[\frac{Z_{\boldsymbol{\theta}_\star}}{k}\sum_{i=1}^{k}\sum_{j=1}^{m}\theta_\star(j)^\zeta f(\boldsymbol{x}_i)1_{J(\boldsymbol{x}_i)=j}\right] - \int_{\mathcal{X}}f(\boldsymbol{x})\pi(d\boldsymbol{x})\right| \\
&= Z_{\boldsymbol{\theta}_\star}\left|\mathbb{E}\left[\frac{1}{k}\sum_{i=1}^{k}\left(\sum_{j=1}^{m}\theta_\star^\zeta(j)f(\boldsymbol{x}_i)1_{J(\boldsymbol{x}_i)=j}\right)\right] - \int_{\mathcal{X}}\left(\sum_{j=1}^{m}\theta_\star^\zeta(j)f(\boldsymbol{x}_i)1_{J(\boldsymbol{x}_i)=j}\right)\varpi_{\widetilde{\Psi}_{\boldsymbol{\theta}_\star}}(\boldsymbol{x})(d\boldsymbol{x})\right|
\end{aligned}
$$

$$(40)$$

Applying the function $\sum_{j=1}^{m}\theta_\star^\zeta(j)f(\boldsymbol{x}_i)1_{J(\boldsymbol{x}_i)=j}$ to Lemma C1 yields

$$
\left|\mathbb{E}\left[\frac{1}{k}\sum_{i=1}^{k}f(\boldsymbol{x}_i)\right] - \int_{\mathcal{X}}f(\boldsymbol{x})\varpi_{\widetilde{\Psi}_{\boldsymbol{\theta}_\star}}(\boldsymbol{x})(d\boldsymbol{x})\right| = \mathcal{O}\left(\frac{1}{k\epsilon} + \sqrt{\epsilon} + \sqrt{\frac{\sum_{i=1}^{k}\omega_i}{k}} + \frac{1}{\sqrt{m}} + \sup_{i\geq k_0}\sqrt{\delta_n(\boldsymbol{\theta}_i)}\right).
$$

$$(41)$$

Plugging (41) into (40) and combining $\mathrm{I}_1$, $\mathrm{I}_{21}$, $\mathrm{I}_{22}$, $\mathrm{I}_3$ and Theorem 1, we have

$$
\left|\mathbb{E}\left[\frac{\sum_{i=1}^{k}\theta_i^\zeta(\tilde{J}(\boldsymbol{x}_i))f(\boldsymbol{x}_i)}{\sum_{i=1}^{k}\theta_i^\zeta(\tilde{J}(\boldsymbol{x}_i))}\right] - \int_{\mathcal{X}}f(\boldsymbol{x})\pi(d\boldsymbol{x})\right| = \mathcal{O}\left(\frac{1}{k\epsilon} + \sqrt{\epsilon} + \sqrt{\frac{\sum_{i=1}^{k}\omega_i}{k}} + \frac{1}{\sqrt{m}} + \sup_{i\geq k_0}\sqrt{\delta_n(\boldsymbol{\theta}_i)}\right),
$$

which concludes the proof of the theorem.

# D  More discussions on the algorithm

## D.1  An alternative numerical scheme

In addition to the numerical scheme used in (6) and (8) in the main body, we can also consider the following numerical scheme

$$
\boldsymbol{x}_{k+1} = \boldsymbol{x}_k - \epsilon_{k+1}\frac{N}{n}\left[1 + \zeta\tau\frac{\log\theta_k\big(\tilde{J}(\boldsymbol{x}_k)\wedge m\big) - \log\theta_k\big(\tilde{J}(\boldsymbol{x}_k)\big)}{\Delta u}\right]\nabla_{\boldsymbol{x}}\widetilde{U}(\boldsymbol{x}_k) + \sqrt{2\tau\epsilon_{k+1}}\boldsymbol{w}_{k+1}.
$$

Such a scheme leads to a similar theoretical result and a better treatment of $\Psi_{\boldsymbol{\theta}}(\cdot)$ for the subregions that contains stationary points.

## D.2  Bizarre peaks in the Gaussian mixture distribution

A bizarre peak always indicates that there is a stationary point of the same energy in somewhere of the sample space, as the sample space is partitioned according to the energy function in CSGLD. For example, we study a mixture distribution with asymmetric modes $\pi(x) = 1/6N(-6,1)+5/6N(4,1)$. Figure S1 shows a bizarre peak at $x$. Although $x$ is not a local minimum, it has the same energy as "-6" which is a local minimum. Note that in CSGLD, $x$ and "-6" belongs to the same subregion.

Figure S1: Explanation of bizarre peaks.

## D.3  Simulations of multi-modal distributions

We run all the algorithms with 200,000 iterations and assume the energy and gradient follow the Gaussian distribution with a variance of 0.1. We include an additional quadratic regularizer $(\|\boldsymbol{x}\|^2 - 7)1_{\|\boldsymbol{x}\|^2 > 7}$ to limit the samples to the center region. We use a constant learning rate 0.001 for SGLD, reSGLD, and CSGLD; We adopt the cyclic cosine learning rates with initial learning rate 0.005 and 20 cycles for cycSGLD. The temperature is fixed at 1 for all the algorithms, excluding the high-temperature process of reSGLD, which employs a temperature of 3. In particular for CSGLD, we choose the step size $\omega_k = \min\{0.003, 10/(k^{0.8} + 100)\}$ for learning the latent vector. We fix 100 partitions and each energy bandwidth is set to 0.25. We choose $\zeta = 0.75$.

## D.4  Extension to the scenarios with high-$\zeta$

In some complex experiments (e.g. computer vision) with a high-loss function, the fixed point $\boldsymbol{\theta}_\star$ can be very close to the vector $(1, 0, ..., 0)$, i.e., the first subregion contains almost all the probability mass, if the sample space is not appropriately partitioned. As a result, estimating $\theta(i)$'s for the high energy subregions can be quite difficult due to the limitation of floating points. If a small value of $\zeta$ is used, the gradient multiplier $1 + \zeta\tau\frac{\log\theta_\star(i) - \log\theta_\star((i-1)\vee 1)}{\Delta u}$ is close to 1 for any $i$ and the algorithm will perform similarly to SGLD, except with different weights. When a large value of $\zeta$ is used, the convergence of $\boldsymbol{\theta}_\star$ can become relatively slow. To tackle this issue, we include a high-order bias item in the stochastic approximation as follows:

$$\theta_{k+1}(i) = \theta_k(i) + \omega_{k+1}\left(\theta_k^\zeta(\tilde{J}(\boldsymbol{x}_{k+1}) + \omega_{k+1}1_{i \geq \tilde{J}(\boldsymbol{x}_{k+1})}\rho)\right)\left(1_{i=\tilde{J}(\boldsymbol{x}_{k+1})} - \theta_k(i)\right), \quad (42)$$

for $i = 1, 2, \ldots, m$, where $\rho$ is a constant. As shown early, our convergence theory allows inclusion of such a high-order bias term. In simulations, the high-order bias term $\omega_{k+1}^2 1_{i \geq \tilde{J}(\boldsymbol{x}_{k+1})}\rho$ penalized more on the higher energy regions, and thus accelerates the convergence of $\boldsymbol{\theta}_k$ toward the pattern $(1, 0, 0, \ldots, 0)$ especially in the early period.

In all computation for the computer vision examples, we set the momentum coefficient to 0.9 and the weight decay to 25, and employed the data augmentation scheme as in Zhong et al. [2017]. In addition, for CSGHMC and saCSGHMC, we set $\omega_k = \frac{10}{k^{0.75}+1000}$ and $\rho = 1$ in (42) for both CIFAR10 and CIFAR100, and set $\zeta = 1 \times 10^6$ for CIFAR10 and $3 \times 10^6$ for CIFAR100.

## D.5 Number of partitions

A fine partition will lead to a smaller discretization error, but it may increase the risk in stability. In particular, it leads to large bouncy jumps around optima (a large negative learning rate, i.e., $\frac{\log \theta(2) - \log \theta(1)}{\Delta u} \ll 0$ in formula (8) may be caused there). Empirically, we suggest to partition the sample space into a moderate number of subregions, e.g. 10-1000, to balance between stability and discretization error.

## Footnotes

*To whom correspondence should be addressed: Faming Liang.