[Reviews · NeurIPS 2020]

Review 1

Summary and Contributions: The paper proposes extension of stochastic gradient Langevin dynamics method for a better optimal point convergence. The main proposed idea is to design an importance sampling like procedure where the utility distribution is piecewise-differentiable over the energy landscape.

Strengths: The general idea of escaping suboptimal solution in stochastic Monte Carlo is interesting and useful to a large audience.

Weaknesses: The problem the paper sets out to solve is not adequately motivated. Notation and abbreviations made the presentation a little cumbersome.

Correctness: Seems correct for the most part. Did not check theorems and proof's.

Clarity: Clear enough.

Relation to Prior Work: Yes.

Reproducibility: Yes

Additional Feedback: Major comments 1) The failure of flat histogram approaches in SD Langevin is not adequately explained? Could you elaborate on why the gradient log probability would vanish in SGLD? 2. It would be highly informative to have a real world data setup that resembles where CSGLD importance is highlighted, similar to the energy landscape and iteration in depicted in Figure 2. 2) In Table 2. ResNet20 did seem to give a comparable result in CIFAR10 and worse result in CIFAR100 with respect to ResNet18 when trained with M-SGD. Could you elaborate a little more on results? More precisely, why would one want to use CSGLD in CIFAR10 and CIFAR100? when M-SGD with tuned hyperparameters can get to a better optimal solution. Minor comments 1. Notation and organization of the equations made the writing a little difficult to follow 2. It would be good to describe histogram flattening method in some detail since the proposed method seems to extend a similar kind of procedure to SGLD. 3. It would make the paper easier to read, if most derivations are moved to the appendix or supplementary material. Instead, the focus should be on the justification for the approach in comparison to earlier methods and on its convergence guarantee.


Review 2

Summary and Contributions: The authors propose an alternative Langevin dynamics sampler with stochastic gradients that uses weights from a flattened density to update the iterates at each step. Synthetic experimental results show the suitability of the method to obtain better point estimates where the energy functions are sharply multimodal. The algorithm is also shown to obtain better rMSE / Bayesian model average numbers on real regression and CV datasets, compared to standard baselines like SGD, SGLD and SGHMC.

Strengths: 1. The weighting of the stochastic gradients using a flattened / piecewise continuous function is interesting, and could help stochastic samplers escape local optima. The flattening technique used is quite intuitive, and it's motivated and explained well. 2. The theoretical study is generally well detailed, although I did not get time to read the supplementary in detail. The treatment in the supp looks quite thorough. 3. The experiments do demonstrate that the method can provide better point estimates in certain sampling situations compared to widely used baselines. The improvements over SGHMC et al with a decaying learning rate look impressive.

Weaknesses: 1. My main concern is that using a flattened surrogate energy in this fashion is suitable for most sampling situations. The main reason is, by construction our iterates are not following the true distribution particularly closely; for example a plot of the samples obtained in the synthetic experiments (figs 2c--d) would look quite different from the original. While this does allow the algorithm to bounce out of local optima, the deviance from the true energy would make samples obtained after convergence to not be super useful. For point estimation situations, we might be able to get away with these samples for cases where the multiple modes of the real energy are sort of symmetric (as in the synthetic Gaussian experiments); it seems that even if we use a 'flattened' energy (can be thought of as lower peaks with higher elevation between them), the original distribution's symmetry would be essentially preserved and the mean / other point estimates would be close enough. But flattening energies with skewed distribution of modes might not be as accurate, as the flattened version might have a mean closer to the 'center' of the space, but the original would be closer to one of the modes near the periphery (am visualizing a simple 2-d space). 2. In a similar vein, I was envisioning a simple extension of SG-MCMC methods where we just do occasional random walks/Brownian motion in the original space (using Gaussian noise and ignoring the energies entirely) using some relatively cheap heuristic to detect if the fancier iterate sequence is stuck in local optima. Something like that would help explore the entire space (triggering some iterations of random walk if heuristic flags a local optima) without changing the underlying energy used in the real Sg-MCMC at each iteration. Wonder how that would compare to a method like the one in the paper. 3. Following from these, it would have been nice if the authors included synthetic experiments with asymmetric modes (one small mode to one side of a 2d space and one taller one on the other side) where the real mean would not be close to the 'center' and thus would not be easily approximable by a flatter version of the energy. Comparisons with something like the approach in 2. above would help understand the main insight of the paper better. 4. Runtime plots in the experiment sections on real data would be nice to have, for example plots of energy/error vs wall clock time / iteration count as provided for the synthetics. 5. Some insight on how to choose \delta_{u} and partition count for real datasets (like the values mentioned on line 173 for synthetic) would be nice to have.

Correctness: Other than the questions mentioned earlier, I do not have any concerns on correctness of the main text.

Clarity: It's generally well written and good to read, but there's some things that could be improved: 1. The connection between using flattened energy and ``AI safety'' mentioned in the intro was not fully clear to me; are the authors implying the lowered informativeness of the surrogate energy acting as kind of an obfuscation in the sampling stage ? Some details on how this makes the sampler 'safer' would be great to have. 2. In a similar vein, the connection between AI safety and convergence speed in lines 23--24 could be better explained. 3. In lines 13--24 the authors mention issues with convergence speed to be the primary dissuading factor for adoption of sg-MCMC methods. I think this point could be cleared up better (for example speed in terms of # iterations vs wall clock time), as to my understanding the main problem is the relatively complicated formulations / additive random noise needed for SGLD/SGHMC as opposed to SGD that makes them more complex to run. 4. The writing in lines 13--19 and 26--29 could be improved. 5. Would be great if authors could make room to include the assumptions 1--6 used in theorems 1 and 2 in the main text. 6. The mention of randomly chosen minibatch size in the CV experiment (lines 235 --236) could be explained better. How does this impact convergence, and/or how do the other/baseline methods do in a similar regime. The subsequent mention of a (fixed?) batchsize of 1000 (line 239) also seems a bit odd. 7. ~ : Sorting the citation sequences would make them nicer to read.

Relation to Prior Work: I think the connections to prior work were well explained. ===== Post rebuttal+discussion: I agree with my fellow reviewers that the paper's contributions/improvements w.r.t existing literature on the topic should be delineated better.

Reproducibility: Yes

Additional Feedback: Please see the sections above. ===== Post rebuttal+discussion: The inclusion of additional experimental results by the authors in the rebuttal is commendable, but some of my other questions (from the Clarity and Weaknesses sections) remain unanswered. I also agree with my fellow reviewers that the paper's contributions/improvements w.r.t existing literature on the topic should be delineated better. Overall I'm inclined to keep my original score; the paper (and the rebuttal) show a lot of promise, but I feel it needs a bit of work to get into clear accept territory. Hopefully our reviews will be helpful to the authors in that regard.


Review 3

Summary and Contributions: The authors propose applying flat histogram algorithm to SGMCMC in order to sampling from multimodal distributions efficiently. The contribution is a new SGMCMC method that works on multimodal distributions. ===== Post rebuttal: I thank the authors for their response, especially the additional experiments which help understand the comparison to previous work. However, the paper still needs more work on the motivation and the connection/comparison with prior work. I also encourage the authors to improve the presentation of the proposed method.

Strengths: The authors provide both theoretical and empirical results which helps to understand the proposed method comprehensively.

Weaknesses: The motivation for the proposed flat histogram algorithm is weak. I’m not convinced by the drawbacks of simulated annealing and parallel tempering that the authors suggested. Specifically, for simulated annealing, it is unclear what it means by “unstable for fast decaying temperatures and not appropriate for statistical inference any more”. Moreover, [1] proposed a simulated annealing based SGMCMC and demonstrated it to work well on deep neural networks (DNNs). Could the authors explain more about this? For parallel tempering, [2] applied it to SGMCMC and obtained good results on DNNs. Therefore, I think the authors missed several related work and should further motivate the use of flat histogram algorithms. In the experiments, the Gaussian mixture example seems to verify the proposed method’s ability of traversing multiple modes. However, the baselines are too weak. I suggest the authors to compare with other SGMCMC methods that work on multimodal distributions, such as SGMCMC with simulated annealing [1], parallel tempering [2] and cyclical stepsize schedule [3]. These comparisons are important for evaluating the proposed method. [1] Bridging the Gap between Stochastic Gradient MCMC and Stochastic Optimization, AISTATS 2016 [2] Non-convex Learning via Replica Exchange Stochastic Gradient MCMC, ICML 2020 [3] Cyclical stochastic gradient MCMC for Bayesian deep learning, ICLR 2020

Correctness: The claims, method and empricial methodology seem correct to me, but they are not well-supported.

Clarity: The presentation needs significant improvement. Flat histogram method is the foundation of the proposed method. However, the introduction of it is too brief so that it is hard to follow for readers who are not familiar with this method. The introduction of the proposed method is also hard to follow. The authors should explain more about the problem of naively applying flat histogram to SGMCMC. For example, it is unclear to me why SGLD has gradient-vanishing problem and how the authors solve it by using a piecewise continuous function.

Relation to Prior Work: It missed several related work (as listed above) and the comparison between the proposed method and these prior work is not discussed.

Reproducibility: Yes

Additional Feedback:


Review 4

Summary and Contributions: The paper proposes an adaptive sampling algorithm to alleviate the local minima trap issue. The proposed algorithm is based on the stochastic gradient Langevin dynamics algorithms. The novelty of the paper is the addition of an adaptive term that modulates the gradient at each iteration according to the energy level of the previous sample. A theoretical analysis for the convergence of the proposed algorithm is provided along with several numerical examples to highlight the efficiency of the proposed algorithm. NOTE: Read the authors response

Strengths: The paper proposes an interesting improvement of the stochastic gradient Langevin dynamics (SGLD) algorithm. The additional term that modulates the gradient at each iteration is constructed such that to avoid the sample trajectory getting stuck in a local minimum. For that, we have that sometimes the algorithm will move towards higher energy regions, in effect opposite to the direction of the gradient. The computational cost of the additional steps is reduced, thus the algorithm remains competitive with SGLD from this point of view. The authors provide both a theoretical analysis and a numerical evaluation of the performances of the proposed algorithm. More so, they also provide intuition for the functioning of the algorithm. Overall, the paper is well written and it is fairly easy to follow along and understand what are the contributions. I belive that overall the proposed algorithm is of interest to the ML community.

Weaknesses: In figure 1a there are some bizarre peaks that show up in regions of high probability when \varsigma increases. Furthermore, when \varsigma=1 instead of having only two energy valleys, we have one between the original valleys and we have that the energy decreases as we move away from the original energy valleys (regions to the left of -6 and to the right of 4). My understanding in such a case is that the samples that will be drawn will not necessarily be representative of the true distribution given that the algorithm will seek the regions of lowest energy. Figure b seems to confirm this understanding as we can see that all energy levels have roughly the same frequency. I'm also believing that the algorithm might even fail to converge in such a case if for example we arive in the region to the left of -6: we are at a very low energy level, so there won't be any influence to drive the trajectory over the peak at -6, thus the gradient will constantly drive the algorithm further away. Consider also the case for \varsigma=0.75, we see that there is also a strange peak that forms at the original energy valley centered around 4. If the algorithm finds itself in the region between 0 and 4 it will most likely remain there, or that is my understanding, thus not provide samples from the correct distribution. The choice of \varsigma seems to play an important part in the performances of the algorithm, yet I feel not enough attention is devoted to how to choose it in practice. Another important aspect that I feel is insufficiently discussed is how to define the energy level partitioning. Is there any intuition on how to perform this operation besides the more the better? Are there any disadvantage in having too many energy levels? Overall, I feel that slightly more attention could have been devoted to the influence of the hyper-parameters on the performances of the proposed algorithm.

Correctness: The claims and the methodology seems to be correct.

Clarity: The paper is generally well written.

Relation to Prior Work: The authors provided a clear explanation of how the proposed algorithm relates to existing algorithms and what is the novelty.

Reproducibility: Yes

Additional Feedback: I believe there are some typos in the formulas and algorithms: - equation 5: is the sum really running from i=1 to m? - algorithm 1, equation 8: shouldn't the rightmost term contain \bm{\omega}_{k+1} instead of \bm{e}_{k+1} In algorithm 1, when the previous sample is in the lowest possible energy state, what do you take as the value of \delta u?

[Author Response · NeurIPS 2020]

We thank all the reviewers for the valuable comments.

**To Reviewer 1:**

Q1. *Details on the vanishing gradient problem in flat-histogram SGLD:* The original step function in formula (4)
leads to $\frac{\partial \log \Psi_\theta(u)}{\partial u} = \frac{1}{\Psi_\theta(u)} \frac{\partial \Psi_\theta(u)}{\partial u} = 0$ almost everywhere. Combining it with (6) leads to $\nabla_x \log \varpi_{\Psi_\theta}(x) = $
$- \left[1 + \zeta\tau \frac{\partial \log \Psi_\theta(u)}{\partial u}\right] \frac{\nabla_x U(x)}{\tau} = -\frac{\nabla_x U(x)}{\tau}$. Therefore, the naive flat-histogram SGLD will behave like SGLD and fail
to converge to the flattened density (2).

Q2. *Advantages of CSGLD over M-SGD*: (i) CSGLD belongs to the class of adaptive biasing force algorithms and
has the potential to exponentially speed-up the computation [1], i.e. the fine-tuned CSGLD should outperform the
fine-tuned M-SGD. (ii) CSGLD also belongs to the class of dynamic importance sampling algorithms and is able to
quantify uncertainty for its estimation and prediction. While M-SGD is an optimization algorithm and produces a point
estimate only.        [1] Long-time convergence of an adaptive biasing force (ABF) method. Nonlinearity. 2008.

**To Reviewer 2:**

Q1. *The working distribution is different from the original one*: CSGLD belongs to the class of dynamic importance
sampling algorithms, for which each sample is generated with an importance weight and they can be used for inference
of the target distribution via a weighted averaging estimator. We have provided theoretical guarantee for the convergence
of the weighted averaging estimator. The samples from the target distribution can also be obtained via an importance
resampling step from the pool of importance samples generated by CSGLD.

Q2. *Asymmetric modes with a heuristic baseline:* Following sec-
tion 4.1, we set $\pi(x) = 1/6N(-6,1) + 5/6N(4,1)$ and included a
baseline called *Heuristic*, which injects a Gaussian noise $N(0,2^2)$
whenever $|\nabla \widetilde{U}(x)| < 0.1$. As shown by Figure (b) of this rebuttal,
*Heuristic* performs quite well in the early period due to the heuristic
random walk helping to escape local traps. However, it leads to a very
large prediction error in the long run as the sampling equilibrium was
broken by *Heuristic*, but the samples were not properly weighted.

(a) Original v.s. trial densities          (b) Estimation errors

Q3. *Discussions on the number of partitions:* A fine partition will
lead to a smaller discretization error, but it increases the risk in stability. In particular, it will lead to large bouncy
jumps around optima (a large negative learning rate, i.e., $\frac{\log \theta(2) - \log \theta(1)}{\Delta u} \ll 0$ in formula (8) will be caused there).
Empirically, we suggest to partition the sample space into a moderate number of subregions, e.g. 10-1000, to balance
between stability and discretization error.

**To Reviewer 3:**

Q1. *Drawbacks of simulated annealing (SA) and replica exchange SGLD (reSGLD)/parallel tempering:* SA can only be
used in optimization, and it might get stuck in a poor local minimum if the temperature decreases too fast. The reSGLD
requires a large correction to reduce the bias, which may lead to a large bias and insignificant accelerations.

Q2. *Missing baselines*: We further compared CSGLD with CyclicalSGLD and reSGLD on an asymmetric mixture
distribution. All algorithms were run $10^7$ iterations. For CSGLD, we set $\widetilde{U}(x) \sim \mathcal{N}(U(x), 4^2)$, the default stepsize
$\alpha = 0.1$ and temperature $T = 1$. For CyclicalSGLD, we set $\alpha_0 = 1$ and 100 cycles and the threshold $\beta = 0.9$; for
reSGLD, we additionally include a high-temperature process with $T = 5$ and a correction of 16. Figure (b) of this
rebuttal shows that CSGLD is inferior to the baselines at the beginning, but it eventually outperforms the baselines as
the learning of $\theta_k$ is mature. We will include the baselines and references in the next version.

Q3. The gradient-vanishing problem in SGLD is not clear: Please refer to our reply to Q1 of Reviewer 1.

**To Reviewer 4:**

Q1. *Comments on bizarre peaks*: A bizarre peak always indicates that there is a local minimum of the same energy in
somewhere of the sample space, as the sample space is partitioned according to the energy function in CSGLD. For
example, Figure (a) of this rebuttal shows a bizarre peak at $x_1$. Although $x_1$ is not a local minimum, it has the same
energy as "-6" which is a local minimum. Note that in CSGLD, $x_1$ and "-6" belongs to the same subregion.

Q2. *In very low density level with no driving forces.* Within the same subregion, CSGLD is reduced to SGLD. Therefore,
in the very low density region, it will not move farther away, but still move toward the high density region.

Q3. *Choices of the partition and $\zeta$*: Regarding the partition, please refer to our response to Q3 of reviewer 2. With
respect to $\zeta$, we suggest to tune $\zeta$ to obtain the largest barrier reductions such as $\zeta = 0.75$ in Figure 1(a) of the paper.

[Meta-Review · NeurIPS 2020]

The paper presents valuable theoretical and empirical evidence for a novel algorithm. However, the reviewers regarding clarity of the contributions and comparisons to existing work. The AC is confident this represents valuable work but was a bit torn about the acceptance decision, as the reviewers point out several important avenues where improvement in the paper is needed.